

# Climatic controls on leaf wax hydrogen isotope ratios in terrestrial and marine sediments along a hyperarid to humid gradient

Nestor Gaviria-Lugo[1], Charlotte Läuchli[2], Hella Wittmann[1], Anne Bernhard[2], Patrick Frings[1], Mahyar Mohtadi[3], Oliver Rach[1], Dirk Sachse[1]

[1]GFZ German Research Centre for Geosciences, Potsdam, Germany
[2]Institute of Geological Sciences, Free University of Berlin, Berlin, Germany
[3]MARUM-Center for Marine Environmental Sciences, University of Bremen, Bremen, Germany

*Correspondence to*: Nestor Gaviria-Lugo (nestgav@gfz-potsdam.de)

**Abstract.** The hydrogen isotope composition of leaf wax biomarkers ($\delta^2H_{wax}$) is a valuable tool for reconstructing continental paleohydrology, as it serves as a proxy for the hydrogen isotope composition of precipitation ($\delta^2H_{pre}$). To yield robust palaeohydrological reconstructions using $\delta^2H_{wax}$ in marine archives, it is necessary to examine the impacts of regional climate on $\delta^2H_{wax}$ and assess the similarity between marine sedimentary $\delta^2H_{wax}$ and the source of continental $\delta^2H_{wax}$. Here, we examined an aridity gradient from hyperarid to humid along the Chilean coast. We sampled sediments at the outlets of rivers draining into the Pacific, soils within catchments and marine surface sediments adjacent to the outlets of the studied rivers and analyzed the relationship between climatic variables and $\delta^2H_{wax}$ values. We find that apparent fractionation between leaf waxes and source water is relatively constant in humid/semiarid regions (average: -121 ‰). However, it becomes less negative in hyperarid regions (average: -86 ‰) as a result of evapotranspirative processes affecting soil and leaf water $^2H$ enrichment. We also observed that along strong aridity gradients, the $^2H$ enrichment of $\delta^2H_{wax}$ follows a non-linear relationship with water content and water flux variables, driven by strong soil evaporation and plant transpiration. Furthermore, our results indicated that $\delta^2H_{wax}$ values in marine surface sediments largely reflect $\delta^2H_{wax}$ values from the continent, confirming the robustness of marine $\delta^2H_{wax}$ records for paleohydrological reconstructions along the Chilean margin. These findings also highlight the importance of considering the effects of hyperaridity in the interpretation of $\delta^2H_{wax}$ values and pave the way for more quantitative paleohydrological reconstructions using $\delta^2H_{wax}$.

## 1 Introduction

The assessment of changes in paleohydrology is crucial to reconstruct a complete picture of paleoclimate. Paleohydrological changes can be inferred with multiple proxies, including pollen analyses, oxygen stable isotopes from foraminifera, hydrogen and oxygen stable isotopes from ice cores, stable isotopes in tree ring cellulose, and leaf wax n-alkanes and their stable carbon and hydrogen isotopes (Dansgaard et al., 1993; Eglinton and Eglinton, 2008; Epstein et al., 1977; Francey and Farquhar, 1982; Lisiecki and Raymo, 2005; Longinelli, 1984; North Greenland Ice Core Project members, 2004; Sachse et al., 2012). Among these proxies, leaf wax derived long chain n-alkanes and their hydrogen isotope ratios have proven particularly useful for



reconstructing changes in continental paleohydrology (Niedermeyer et al., 2010; Pagani et al., 2006; Rach et al., 2014; Schefuß et al., 2005; Tierney et al., 2008; Collins et al., 2017), because of their long-term preservation potential, their source specificity and their ubiquitous presence in sedimentary archives.

Leaf wax hydrogen isotope ratios (expressed here as $\delta^2H_{wax}$) reflect the hydrogen isotopic composition of local precipitation ($\delta^2H_{pre}$). Studies of modern plants, soils and lake surface sediments have shown that $\delta^2H_{wax}$ and $\delta^2H_{pre}$ are highly correlated, and that $\delta^2H_{pre}$ is generally the primary control on $\delta^2H_{wax}$ (Polissar and Freeman, 2010; Rao et al., 2009; Sachse et al., 2004,
2006, 2012; Sessions et al., 1999). Hence, $\delta^2H_{wax}$ values are considered a high fidelity recorder of $\delta^2H_{pre}$, and consequently of the processes affecting $\delta^2H_{pre}$ values. In mid to high latitudes $\delta^2H_{pre}$ values are controlled by distance from the coastline (continentality/rainout effect) and changes in temperature of the air masses, in turn controlled by elevation and latitude effects (Bowen et al., 2019; Craig, 1961; Dansgaard, 1964; Gat, 1996; Rozanski et al., 1993). In tropical regions, $\delta^2H_{pre}$ values primarily respond to rainfall amount (Hoffmann et al., 2003; Kurita et al., 2009; Vuille and Werner, 2005). Beyond this,
moisture source is another important factor dictating the isotopic signature of precipitation (Tian et al., 2007; Uemura et al., 2008; Vimeux et al., 1999, 2001).

Before being incorporated into leaf waxes, $\delta^2H_{pre}$ values are modified by biosynthetic processes inside plants and ecohydrological processes like evapotranspiration. The net effect of these fractionations is such that $\delta^2H_{wax}$ is significantly depleted in deuterium relative to the source water $\delta^2H_{pre}$, with the depletion commonly referred to as net or apparent
fractionation (expressed here as $\epsilon_{wax/pre}$) (Cernusak et al., 2016; Dawson et al., 2002; Kahmen et al., 2013b, a; Smith and Freeman, 2006). Commonly estimated values of $\epsilon_{wax/pre}$ generally average around -120 ‰ (Sachse et al., 2012; Chen et al., 2022). It has been identified that $\epsilon_{wax/pre}$ values become less negative as aridity intensifies (Douglas et al., 2012; Feakins and Sessions, 2010; Garcin et al., 2012; Goldsmith et al., 2019; Herrmann et al., 2017; Li et al., 2019; Polissar and Freeman, 2010; Smith and Freeman, 2006). However, prior research has primarily been confined to regions with an aridity index ≥ 0.05 i.e.,
arid, semiarid, dry-subhumid, and humid regions, where aridity index is defined as the ratio of mean annual precipitation to potential evapotranspiration (after UNEP (1997)), leaving hyperarid zones (aridity index < 0.05) understudied. To improve our understanding of the climatic controls on $\delta^2H_{wax}$ and $\epsilon_{wax/pre}$, it is necessary to examine the impact of hyperaridity and the extent to which hydrological parameters influence $\delta^2H_{wax}$. This information is necessary to maximize the accuracy of paleohydrological reconstructions and eventually develop them into quantitative tools.

In addition to these knowledge gaps, $\delta^2H_{wax}$ values from sedimentary archives may or may not reflect the entire source region due to filtering during sediments transit through sedimentary systems. Häggi et al. (2016) showed that $\delta^2H_{wax}$ from suspended sediments in the Amazon River and $\delta^2H_{wax}$ from marine surface sediments have equivalent values limited to the Amazon freshwater plume, with $\delta^2H_{wax}$ values disagreeing beyond this area. Along the Italian Adriatic coast, marine and terrestrial sediments display equivalent $\delta^2H_{wax}$ values on the semiarid side of the peninsula, but not in the humid regions (Leider et al.

2013). Vogts et al. (2012) showed that δ13Cwax values from marine sediments correlate with δ13Cwax values from terrestrial plants along a humid to arid transect offshore SW Africa. However, the current understanding of the possible filter effects on leaf wax proxy data within sedimentary systems from source to sink remains incomplete. Hence, paired marine-continental sampling approaches would represent a step towards addressing these issues.

      In this study, we analyze leaf wax n-alkanes and their $\delta^2H_{wax}$ values in marine surface sediments, modern soils, and river
sediments along a strong aridity gradient in Chile. The gradient spans from the humid to the hyperarid zones and encompasses a range of annual mean precipitation values from 6 to 2300 mm y-1 (Fig. 1). We examine how climatic factors (evapotranspiration, precipitation, aridity, relative humidity, soil moisture, temperature) impact $\delta^2H_{wax}$ and $\varepsilon_{wax/pre}$, and assess the similarity of $\delta^2H_{wax}$ values among soils, riverine and marine sediments across the aridity zones of the aridity gradient. We provide a better and quantitative understanding of the effects of aridity on $\delta^2H_{wax}$ values and a foundation for the application
of $\delta^2H_{wax}$ in paleohydrologic reconstructions along aridity gradients, in particular along the Chilean aridity gradient.

## 2 Methods

### 2.1 Study area and sampling strategy

Here we study a gradient from hyperarid to humid along Chile, using soils, river sediments and marine surface sediments (Fig. 1). During March-April 2019 we sampled topsoils (upper 5 cm, n=12) and riverbed sediments (n=26) from catchments draining
to the Pacific Ocean. Three small sub-catchments nested in three of the major catchments were also sampled. Marine core top sediments (1-2 cm core depth, 29 sites, Table 1) were provided by the MARUM core repository. Marine samples used in this study were recovered during expeditions SO-102 and SO-156 of the RV SONNE using a multicorer (Hebbeln, 1995, 2001).

### 2.2 Analytical methods

### 2.2.1 Sample preparation

Prior to extraction of the organic compounds, samples from river and soil sediment were dried for 48h in an oven at 60 °C. Any visible plant material was carefully extracted using steel tweezers, and the samples were sieved to <32 $\mu$m. Marine surface sediments were freeze-dried for 48h and homogenized using an agate mortar.

### 2.2.2 Lipid extraction and chromatography

Organic lipids were extracted from ~20 g of sediment and purified using the manual solid phase extraction (SPE) procedure
described in Rach et al. (2020). In brief, a Dionex Accelerated Solvent Extraction system (ASE 350, ThermoFisher Scientific) using a dichloromethane (DCM):Methanol (MeOH) mixture (9:1, v/v) at 100 °C, 103 bar pressure with two extraction cycles



(20 min static time) was used to extract total lipid extracts (TLE). Those were collected in a combusted glass vial and then completely evaporated under a stream of nitrogen. All samples were spiked with 10 µg of an internal standard (5α-androstane) and the TLE was separated into aliphatic, aromatic and alcohol/fatty acid fractions by solid phase extraction in 8 ml glass columns filled with 2 g silica gel using Hexane, Hexane:DCM (1:1) and DCM:Methanol (1:1) as solvents. The aliphatic fraction was desulphurized by elution through activated copper powder, and then additionally cleaned over silver nitrate coated silica gel.

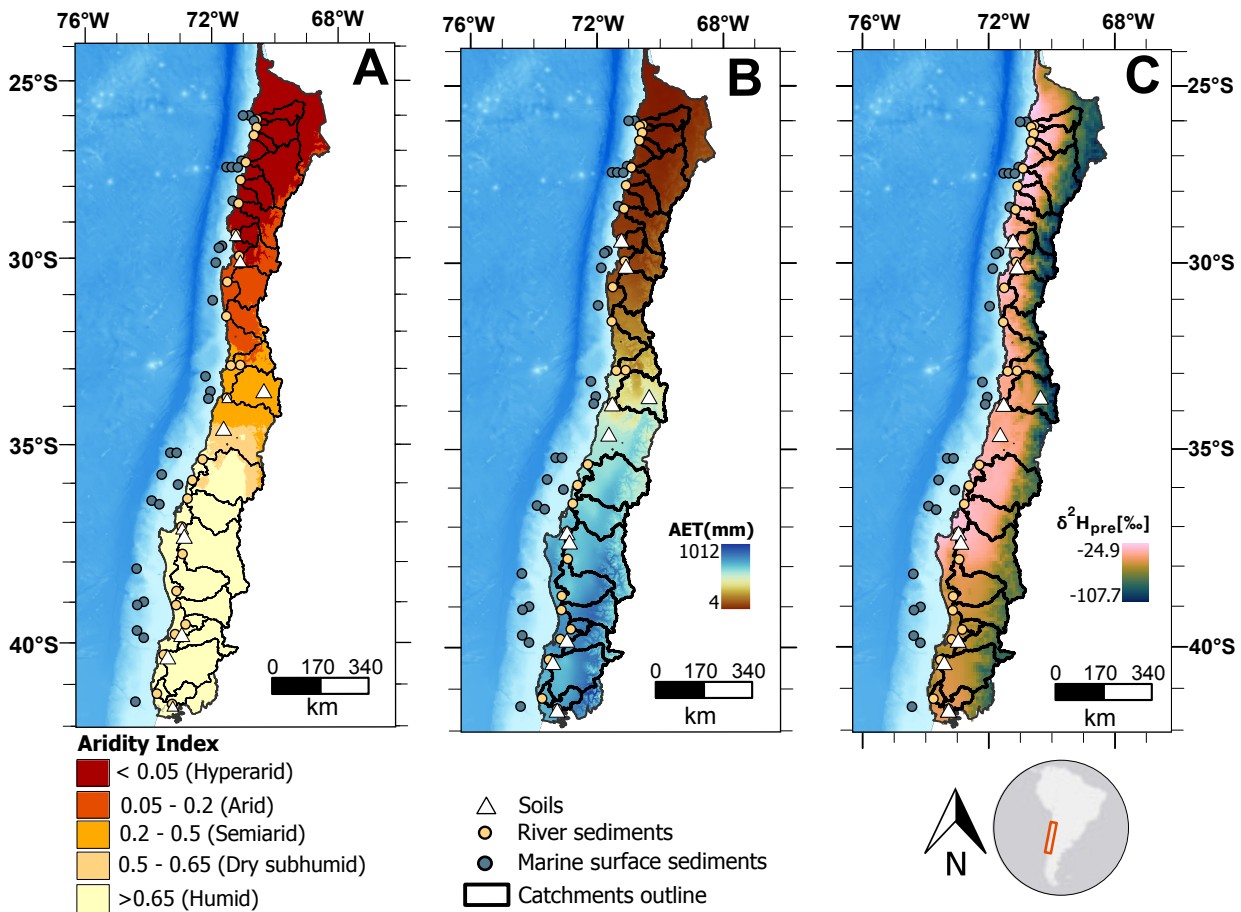

**Figure 1:** Maps of climatic conditions and sampling locations along the studied gradient. (A) Aridity index (dimensionless quantity) with sampling locations and studied catchments. Aridity index data is from the Global Aridity Index and Potential Evapotranspiration Climate Database v3 of Trabucco and Zomer (2022) and aridity zone classification follows the classification proposed by UNEP (1997). (B) Long term (1958-2015) mean annual actual evapotranspiration (AET; mm y$^{-1}$) from the TerraClimate dataset of Abatzoglou et al. (2018). (C) Mean annual hydrogen isotopic composition of precipitation ($\delta^2H_{pre}$, in ‰), from The Online Isotopes in Precipitation Calculator, version 3.1.



Identification and quantification of n-alkanes was performed on an Agilent 7890A gas chromatograph (GC) coupled to a flame ionization detector (FID) and to an Agilent 5975C mass spectrometer (MS). The GC was equipped with a 30 m Agilent DB-5MS UI column (0.25 µm film thickness, 25 mm diameter). Quantification results were normalized to their initial dry weight and are reported as µg/g dry weight.

### 2.2.3 Hydrogen isotope analysis

Stable hydrogen isotope ratios ($\delta^2$H) from the n-alkanes were measured in the separated aliphatic fractions using a Trace GC 1310 (ThermoFisher Scientific) connected to a Delta V plus Isotope Ratio Mass Spectrometer (IRMS) (ThermoFisher Scientific). The GC was equipped with a 30 m Agilent DB-5MS UI column (0.25 µm film thickness, 25 mm diameter). n-alkane $\delta^2$H values were determined by duplicate measurements. We followed the same GC oven program as described in Rach et al. (2014). The H3+ factor was measured before each sequence and averaged $2.82 \pm 0.14$ mV (n=6) over a period of 5 weeks.

To correct and transfer to the VSMOW scale an n-alkane standard-mix A6 ($n$-C$_{16}$ to $n$-C$_{30}$) with known $\delta^2$H values obtained from A. Schimmelmann (Indiana University) was measured before and after the samples.

### 2.3 *n*-alkane indices

To assess variations in the n-alkane distributions along the Chilean gradient, we calculated the Carbon Preference Index (CPI) and the Average Chain Length (ACL) indices. The CPI measures the relative abundances of odd vs. even-numbered n-alkanes,

using the concentrations of odd and even numbered n-alkane chains from $n$-C$_{25}$ to $n$-C$_{35}$ following Marzi et al. (1993):

$$CPI = \frac{(C_{25}+C_{27}+C_{29}+C_{31}+C_{33})+(C_{27}+C_{29}+C_{31}+C_{33}+C_{35})}{2(C_{26}+C_{28}+C_{30}+C_{32}+C_{34})} \tag{1}$$

The ACL value is the weighted average of the various carbon chain lengths in a sample:

$$ACL = \frac{(25\times C_{25})+(27\times C_{27})+(29\times C_{29})+(31\times C_{31})+(33\times C_{33})+(35\times C_{35})}{C_{25}+C_{27}+C_{29}+C_{31}+C_{33}+C_{35}} \tag{2}$$

### 2.4 Global $\delta^2$H$_{wax}$ data compilation

The compiled dataset is accessible on the Gaviria-Lugo et al. (2022) data publication and can be correspondently used and cited. We used the previously published $\delta^2$H$_{wax}$ datasets of soils and lake sediments of Ladd et al. (2021) and Chen et al. (2022), but significantly expanded these compilations with newer datasets reporting $\delta^2$H$_{wax}$ for both $n$-C$_{29}$ and $n$-C$_{31}$, In total, our compilation includes data from 26 peer-reviewed publications, with 750 and 663 $\delta^2$H$_{wax}$ values for $n$-C$_{29}$ and $n$-C$_{31}$, respectively

(Table S1).



## 2.5 Remote sensing data and GIS methods

### 2.5.1 δ²H of precipitation

$\delta^2H_{pre}$ was extracted from the grids produced by Bowen and Revenaugh (2003) which are publicly available at the Online Isotopes in Precipitation Calculator webpage (OIPC, The Online Isotopes in Precipitation Calculator, version 3.1). We used the raster grid of annual averaged $\delta^2H_{pre}$ data from the OIPC to calculate the $\delta^2H_{pre}$ values for each location in our study area and from the global compilation. The raster grid data was provided at a resolution of ~9 km, with each pixel corresponding to an area of ~81 km². To assess the accuracy of OIPC $\delta^2H_{pre}$ values at our sampling sites, we compare the values predicted by the OIPC, extracted on a monthly basis, with a dataset comprising 923 measured data points of $\delta^2H_{pre}$. This dataset was obtained from the International Atomic Energy Agency (IAEA/WMO, 2023), collected as part of the Global Network of Isotopes in Precipitation (GNIP) program at 9 long-term monitoring stations located along the different aridity zones of continental Chile.

### 2.5.2 Climatic parameters

Hydrological variables were obtained from the TerraClimate dataset (Abatzoglou et al., 2018) as long term annual averages for the period between 1958-2019 at ~4.5 km spatial resolution, with each pixel representing ~20 km². Variables obtained from the dataset were mean annual precipitation (MAP), actual evapotranspiration (AET), soil moisture (SM), actual vapor pressure (VAP) and vapor pressure deficit (VPD); the latter two were used to derive relative humidity (RH) using Eq. (3):

$$RH(\%) = \frac{VAP}{VAP + VPD} \times 100 \tag{3}$$

The TerraClimate dataset was accessed and analyzed using the cloud computing capabilities publicly available via Google Earth Engine. Aridity index (AIdx) data was accessed from the Consultative Group of the International Agricultural Research Consortium for Spatial Information (CGIARCSI) Global-Aridity Index dataset (Trabucco and Zomer, 2022) that integrates aridity over the period between 1970-2000 at a resolution of ~1 km, with each pixel corresponding to ~1 km². The WorldClim dataset (Fick and Hijmans, 2017) was used to derive the mean annual temperature (MAT) and annual average of daily maximum temperature (MaxT), based on data from 1970-2000 with a resolution of ~1 km, corresponding to ~1 km² per pixel.

### 2.5.3 Vegetation cover

Fractional land cover data was obtained from Collection 2 of the Copernicus Global Land Cover layers (Buchhorn et al., 2020) via Google Earth Engine. We extracted mean values for the fraction of trees, shrubs, grasses, crops, and barren land for the period between 2015-2019 at a 100 m resolution, with each pixel representing 0.01 km². We used land cover fractions of vegetation obtained for each site to derive values of the fraction of herbaceous vegetation and woody vegetation. To derive the woody fraction of the vegetation, we summed the values of trees and shrubs land cover and divided this sum by the sum of all





the vegetation fractions (trees, shrubs, grasses, crops). The values of the herbaceous fraction of the vegetation were derived

summing the values of grasses and crops and dividing this sum by the sum of all the vegetation fractions (trees, shrubs, grasses, crops).

### 2.5.4 Spatial analysis

Catchments were defined upstream of the sampling points of river sediments using the *drainagebasins* function from the MATLAB-based software TopoToolbox 2 (Schwanghart and Scherler, 2014). The digital elevation model (DEM) used for the

drainage basin definition in this study was the freely available Copernicus WorldDEM with a resolution of 30 m (Fahrland et al., 2020). For each catchment area, we calculated a mean value of $\delta^2H_{pre}$, the climatic parameters, and vegetation cover fractions (Table S6) of all grid cells within the catchment. Soil samples were considered as points, in this case we extracted the values of $\delta^2H_{pre}$, the climatic parameters, and vegetation cover fractions from the pixel containing the sampling point (Table S6). For the compiled global dataset all soil samples were also treated as points, and drainage basins were defined for lake

sediment samples using the same procedure as for the Chilean river samples. Values for $\delta^2H_{pre}$ and all climatic parameters were retrieved using the same procedures and the same sources as for our sampling sites in Chile (Table S1).

## 2.6 Statistical and mathematical methods

### 2.6.1 $\delta^2H_{wax}$ vs $\delta^2H_{pre}$ regression and analysis of residuals

Using the R programming language, we conducted a linear regression between the global compilation of $\delta^2H_{wax}$ values and

their corresponding $\delta^2H_{pre}$ values retrieved from the OIPC. This global regression serves as an indicator of the expected relationship between $\delta^2H_{wax}$ and $\delta^2H_{pre}$. To assess the influence of fractionation processes on $\delta^2H_{wax}$ along the Chilean gradient, we calculated the residuals between measured $\delta^2H_{wax}$ and predictions from the global regression for our sampling sites. As $\delta^2H_{pre}$ is assumed to be the primary determinant of $\delta^2H_{wax}$, any substantial deviations from the global regression may suggest the presence of additional fractionation processes.


### 2.6.2 Statistical test of the differences (Kruskal-Wallis test)

We used the Kruskal-Wallis test to investigate the statistical significance of differences in the median values between independent categorical groups. We explored the significance of difference in $\delta^2H_{wax}$ values among the aridity zones, and among the different sediment types analyzed in each aridity zone. This robust non-parametric test allows for comparison of

groups with dissimilar variances and does not require normally distributed data within groups. (Kruskal and Wallis, 1952). We



performed the test using the function ***kruskal.test()*** of the *stats* package version 4.2.1 from the R programming language (R Core Team, 2022).

### 2.6.3 Leaf wax hydrogen isotope fractionation

We calculated the difference between $\delta^2H_{pre}$ values as source water and $\delta^2H_{wax}$ values to assess the climatic parameters that control $\delta^2H_{wax}$ along the gradient. This difference is referred to as the net or apparent fractionation, as it encompasses several fractionation processes, such as the evaporation and enrichment of soil water, the enrichment of leaf water through transpiration and the biosynthesis of lipids (Sessions et al., 1999; Smith and Freeman, 2006; Farquhar et al., 2007; Sachse et al., 2012). We used the isotopic fractionation factor (α) instead of the apparent fractionation in per mil (ε ‰) to derive accurate regressions and to be mathematically consistent according to the theory of compositional data analysis (Aitchison, 1982, 1986), but reported all values as ε (in ‰) for the sake of comparison with other studies. Since α is the ratio of the isotopic composition of two substances, it permits application of the log-ratio approach when studying the factors affecting fractionation. Using log-ratios is the preferred approach to yield mathematically and statistically robust regression models for compositional data (Aitchison, 1982, 1986)(Ramisch et al., 2018; Weltje et al., 2015).

To obtain the fractionation factor α, delta values were transformed back from the per mil scale using Eq. (4) and Eq. (5). The apparent fractionation is related to the isotopic fractionation factor through Eq. (6). For comparison with existing literature data, apparent fractionation values in per mil can be obtained multiplying Eq. (6) by 1000.

$$\delta_i = \frac{\delta_i\text{‰}}{1000} \tag{4}$$

$$\alpha_{wax/pre} = \frac{\delta_{wax}+1}{\delta_{pre}+1} \tag{5}$$

$$\varepsilon_{wax/pre} = \alpha_{wax/pre} - 1 \tag{6}$$

To explore the environmental controls on fractionation, we fitted ordinary least squares regression models between the natural log of the isotopic fractionation factor and the natural log of the environmental parameters of interest. The type of linear model obtained is represented by Eq. (7). To back-transform this type of model from the log space to the original space, we applied an exponential function to both sides of Eq. (7) to obtain a model in the form of Eq. (8):

$$ln\left(\alpha_{wax/pre}\right) = a * ln(X) + b \tag{7}$$

$$\alpha_{wax/pre} = X^a * e^b \tag{8}$$





### 2.6.4 Soil and leaf water enrichment models

To mechanistically understand how evapotranspirative $^2$H enrichment with respect to source water regulates $\delta^2H_{wax}$, it is advantageous to model the effect of this process considering its two primary components, 1) soil water $^2$H enrichment due to evaporation and 2) leaf water $^2$H enrichment due to transpiration. To model soil water $^2$H enrichment we used a simplified Craig-Gordon model based on the modifications of Gat (1995) and followed the parameterization of Smith and Freeman (2006). The soil water is treated as a through-flow reservoir at hydrologic and isotopic steady state. The $^2$H enrichment ($\Delta^2H_{SW}$) is predicted for each site using Eq. (9) based on relative humidity (RH), and the ratio between precipitation (MAP) and evaporated soil water (SEv). RH and MAP data come from the retrieved values for each site (Section 2.5.2; Table S6). The equilibrium isotope fractionation factor between vapor and liquid ($\varepsilon^*$) is calculated as a function of temperature following the empirical relation derived by Horita and Wesolowski (1994).

$$\Delta^2H_{SW} = \frac{(1-RH)\times(\varepsilon^*+12.5)}{RH+\left[(1-RH)\times\left(\frac{MAP}{SEv}\right)\right]} \tag{9}$$

The value of SEv is derived from AET from Eq. (10), in which the soil evaporation factor ($f_e$) reflects the contribution of soil evaporation to AET. This factor $f_e$ varies along the aridity gradient.

$$SEv = f_e \times AET \tag{10}$$

It is known that the contribution of (soil) evaporation to AET is higher in arid regions compared to humid regions (Lawrence et al., 2007; Zhang et al., 2016). According to Lehmann et al. (2019), in arid regions, only 13% of the precipitated water is shielded from evaporation, and this percentage decreases to 2.3% in hyperarid regions. Based on these values, which consider evaporation not only from soils but also from water bodies, we assume conservative $f_e$ values of 0.8 for arid regions and 0.9 for hyperarid regions. The present study does not model the contribution of soil evaporation to $^2$H enrichment in humid regions, as it is typically less significant than transpiration in such regions (Schlesinger and Jasechko, 2014; Zhang et al., 2016).

Leaf water enrichment is modeled using the Péclet modified Craig-Gordon model proposed by Kahmen et al. (2011) for $\delta^{18}$O values, adjusted for $\delta^2$H values as in Rach et al. (2017). Based on Eq. (11) this model first estimates leaf water $^2$H enrichment ($\Delta^2H_{LW}$) considering $\varepsilon^*$ (defined as for the soil evaporation model), the kinetic isotope fractionation during water vapor diffusion from the leaf intercellular air space to the atmosphere ($\varepsilon_k$), the $^2$H enrichment of water vapor relative to source water ($\Delta^2H_{WV}$), and the ratio of atmospheric vapor pressure and leaf internal vapor pressure ($e_a/e_i$).

$$\Delta^2H_{LW} = \varepsilon^* + \varepsilon_k + (\Delta^2H_{WV}-\varepsilon_k)\frac{e_a}{e_i} \tag{11}$$

$\varepsilon_k$ as in Eq. (12) is dependent on two parameters that are considered constant: leaf boundary layer resistance ($r_b = 1$ mol m$^{-2}$ s$^{-1}$) and stomatal conductance ($g_s = 0.1$ mol$^{-1}$ m$^2$ s) which is formulated as stomatal resistance $r_s=1/g_s$.




$$\varepsilon_k = \frac{(16.4 \times r_s) + (10.9 \times r_b)}{r_s + r_b} \tag{12}$$

The magnitude of $\Delta^2H_{WV}$ (Eq. 13) is considered the same as $\varepsilon^*$, as demonstrated by long-term observations in temperate regions (Jacob and Sonntag, 1991).

$$\Delta^2H_{WV} = -\varepsilon^* \tag{13}$$

Leaf internal vapor pressure ($e_i$; Eq. 14) is dependent on leaf temperature ($T_{leaf}$) which is considered to be the same as air
temperature ($T_{air}$) over decadal timescales of sedimentary integration (Rach et al., 2017). Saturation vapor pressure ($e_{sat}$; Eq. 15) depends on $T_{air}$ and atmospheric pressure ($e_{atm}$) at each site. With the value of $e_{sat}$ we calculate the atmospheric vapor pressure ($e_a$; Eq. 16).

$$e_i = 6.13753 \times \exp\left(T_{air} \times \frac{18.564 - \frac{T_{air}}{254.4}}{T_{air} + 255.57}\right) \tag{14}$$

$$e_{sat} = \frac{1.0007 + 3.46 \times e_{atm}}{1\,000\,000} \times 6.1121 \times \exp\left(\frac{17.502 \times T_{air}}{240.97 + T_{air}}\right) \tag{15}$$

$$e_a = \frac{RH}{100} \times e_{sat} \tag{16}$$

The final component of the model involves the utilization of the Péclet number ($\wp$) to adjust for physiological factors that can influence leaf water enrichment. The calculation of $\wp$ entails the preliminary estimation of the transpirational water flux (Tr; Eq. 17) and water diffusivity ($D_{iff}$; Eq. 18).

$$Tr = \frac{e_i + e_a}{r_b + r_s} \tag{17}$$

$$D_{iff} = 10^{-8} \exp\left(-0.7 + \frac{1729}{T[K]} - \frac{586977}{T[K]^2}\right) \tag{18}$$

For the calculation of $\wp$ (Eq. 19), path length ($L_M = 15$ mm) and the molar concentration of water ($C = 5.56 \times 10^4$ mol m$^{-3}$) are considered constant, following Kahmen et al. (2011). After obtaining $\wp$, the final Peclét corrected leaf water $^2$H enrichment values ($\Delta^2H_{LWP}$; Eq. 20) can be calculated.

$$\wp = \frac{L_M \times Tr}{C \times D_{iff}} \tag{19}$$

$$\Delta^2H_{LWP} = \frac{\Delta^2H_{LW}(1 - e^{-\wp})}{\wp} \tag{20}$$



# 3 Results

## 3.1 Concentration and distribution of leaf wax n-alkanes along the Chilean aridity gradient

### 3.1.1 Concentration and distribution of n-alkanes in river sediments and soils

In both soils and riverine sediments *n*-alkane distributions exhibited a strong predominance of odd over even chain length *n*-alkanes with a CPI between 6.5 and 26.4 (Table 1). Total concentrations of *n*-alkanes in riverine sediments ranged from 0.2 $\mu g\ g^{-1}$ sediment dry weight (dw) to 24.3 $\mu g\ g^{-1}$ dw. Total concentration of *n*-alkanes in soils ranged from 9.2 $\mu g\ g^{-1}$ dw to 93.4 $\mu g\ g^{-1}$ dw. Concentrations of *n*-alkanes in both riverine sediments and soil samples were lowest in the hyperarid zone and highest in the humid zone (Table 1). ACL varied along the aridity gradient (Table 1), ranging from 28.6 to 30.9; the highest

ACL values were found in the hyperarid zone and lowest in the humid zone (Table 1). The most prevalent *n*-alkane homologues were *n*-$C_{31}$ and *n*-$C_{29}$. In the hyperarid and arid zones *n*-$C_{31}$ was the most common, while in the semiarid and humid zones *n*-$C_{29}$ predominated (Table 1 and Table S4).

**Table 1.** *n*-alkanes concentration, *n*-alkanes indices, $\delta^2 H_{pre}$ and $\delta^2 H_{wax}$

| Sampling Site | Sediment type | IGSN | Aridity Zone | Sampling Location | | CPI | ACL | C29/ (C29+C31) | Concentration Total n-alkanes | $\delta^2 H$ precipitation | $\delta^2 H$ n-$C_{29}$ | $\delta^2 H$ n-$C_{31}$ |
| --- | --- | --- | --- | --- | --- | --- | --- | --- | --- | --- | --- | --- |
| | | | | Lat (°) | Long (°) | | | | ug g$^{-1}$ sediment dry weight | [ ‰ ] | [ ‰ ] | [ ‰ ] |
| Quebrada Salitrosa | Riverine | GFNG10013 | Hyperarid | -26.1116 | -70.5561 | 12.9 | 30.9 | 0.30 | 1.4 | -29.1 ± 2.6 | -89.9 ± 0.6 | -114.3 ± 0.6 |
| Rio Pan de Azucar | Riverine | GFNG10000 | Hyperarid | -26.1487 | -70.6527 | 12.1 | 29.8 | 0.45 | 0.3 | -50.8 ± 3.2 | -126.2 ± 2.2 | -139.0 ± 3.9 |
| Rio Salado | Riverine | GFNG10012 | Hyperarid | -26.3366 | -70.5699 | 6.5 | 29.9 | 0.35 | 0.5 | -47.8 ± 3.1 | -138.8 ± 0.5 | -150.7 ± 3.8 |
| Quebrada Pto Flamenco | Riverine | GFNG1000S | Hyperarid | -26.5626 | -70.6589 | 18.0 | 30.5 | 0.31 | 1.3 | -40.2 ± 2.9 | -118.7 ± 0.6 | -132.1 ± 11.8 |
| Rio Copiapo | Riverine | GFNG10014 | Hyperarid | -27.3281 | -70.9136 | 26.4 | 30.9 | 0.24 | 2.9 | -62.6 ± 5.0 | -148.9 ± 6.1 | -159.6 ± 2.6 |
| Quebrada Totoral | Riverine | GFNG1000T | Hyperarid | -27.8321 | -71.0816 | 11.0 | 30.3 | 0.37 | 2.3 | -48.9 ± 3.3 | -151.5 ± 2.0 | -153.8 ± 1.8 |
| Rio Huasco | Riverine | GFNG10015 | Arid | -28.4873 | -71.1486 | 10.7 | 29.9 | 0.47 | 1.3 | -69.5 ± 5.2 | -158.2 ± 9.0 | -158.5 ± 9.3 |
| Quebrada Los Choros | Riverine | GFNG1000U | Hyperarid | -29.3302 | -71.2364 | 8.6 | 30.0 | 0.42 | 0.2 | -54.7 ± 2.8 | -151.6 ± 14.5 | -152.1 ± 14.5 |
| Rio Elqui | Riverine | GFNG10016 | Arid | -29.9459 | -71.1293 | 15.7 | 30.2 | 0.44 | 1.1 | -70.2 ± 5.2 | -154.7 ± 11.0 | -163.3 ± 10.8 |
| Rio Limari | Riverine | GFNG10017 | Arid | -30.6585 | -71.5072 | 17.5 | 29.9 | 0.48 | 10.9 | -59.7 ± 3.4 | -165.4 ± 2.8 | -170.5 ± 1.9 |
| Rio Choapa | Riverine | GFNG10018 | Arid | -31.5922 | -71.5382 | 16.9 | 30.1 | 0.48 | 1.1 | -59.7 ± 2.9 | -158.7 ± 2.3 | -160.5 ± 0.2 |
| Quebrada La Campana | Riverine | GFNG10019 | SemiArid | -32.9082 | -71.0994 | 9.3 | 29.4 | 0.54 | 14.3 | -53.4 ± 1.3 | -170.0 ± 1.7 | -165.4 ± 5.8 |
| Rio Aconcagua | Riverine | GFNG1000V | SemiArid | -32.9196 | -71.3959 | 19.8 | 29.8 | 0.53 | 3.3 | -65.4 ± 3.2 | -168.6 ± 4.5 | -170.0 ± 2.7 |
| Rio Maipo | Riverine | GFNG1001A | SemiArid | -33.7738 | -71.5272 | 8.9 | 29.7 | 0.51 | 3.1 | -63.3 ± 2.8 | -167.2 ± 6.8 | -173.1 ± 6.8 |
| Rio Maule | Riverine | GFNG1000W | Humid | -35.3882 | -72.2978 | 7.9 | 28.8 | 0.65 | 0.4 | -51.5 ± 1.8 | -155.2 ± 0.7 | -156.8 ± 7.3 |
| Rio Chovellen | Riverine | GFNG1001C | Humid | -35.9254 | -72.6230 | 10.5 | 29.0 | 0.59 | 7.2 | -42.7 ± 0.9 | -158.0 ± 3.0 | -158.6 ± 1.7 |
| Rio Itata | Riverine | GFNG1000X | Humid | -36.3999 | -72.7762 | 7.1 | 29.1 | 0.63 | 15.4 | -47.3 ± 1.2 | -175.4 ± 0.2 | -169.6 ± 0.0 |
| Rio BioBio | Riverine | GFNG1001D | Humid | -37.1131 | -72.9647 | 10.9 | 29.1 | 0.65 | 1.5 | -54.9 ± 1.7 | -158.1 ± 1.8 | -159.0 ± 1.4 |
| Estero Los Gringos | Riverine | GFNG1000Y | Humid | -37.8035 | -72.9371 | 19.9 | 29.6 | 0.40 | 16.8 | -57.4 ± 2.3 | -173.6 ± 0.1 | -167.0 ± 0.4 |
| Rio Imperial | Riverine | GFNG1000Z | Humid | -38.7287 | -73.1267 | 9.4 | 29.5 | 0.54 | 4.4 | -54.5 ± 0.9 | -163.4 ± 1.4 | -165.8 ± 6.3 |
| Rio Tolten | Riverine | GFNG1001B | Humid | -39.0747 | -73.1397 | 12.3 | 29.1 | 0.62 | 8.4 | -61.8 ± 1.2 | -179.1 ± 0.2 | -170.2 ± 4.6 |
| Rio Cruces | Riverine | GFNG10010 | Humid | -39.5493 | -72.8419 | 8.8 | 29.1 | 0.58 | 24.3 | -57.3 ± 0.5 | -160.4 ± 6.1 | -165.2 ± 4.3 |
| Rio CalleCalle | Riverine | GFNG1001E | Humid | -39.7902 | -73.1749 | 9.7 | 29.4 | 0.54 | 9.9 | -63.5 ± 1.0 | -170.0 ± 2.3 | -169.7 ± 8.1 |
| Rio Bueno | Riverine | GFNG1001F | Humid | -40.2900 | -73.5354 | 7.7 | 29.3 | 0.52 | 2.0 | -61.1 ± 1.0 | -165.1 ± 1.0 | -162.1 ± 2.5 |
| Rio Llico | Riverine | GFNG10011 | Humid | -41.2269 | -73.7530 | 11.8 | 29.3 | 0.51 | 3.3 | -54.8 ± 1.6 | -171.2 ± 5.0 | -167.6 ± 5.2 |
| Rio Maullin | Riverine | GFNG1001G | Humid | -41.4740 | -73.2726 | 8.1 | 29.3 | 0.49 | 9.7 | -53.8 ± 1.4 | -168.4 ± 9.1 | -168.8 ± 3.2 |
| Choros | Soils | GFNG1000F | Hyperarid | -29.3332 | -71.2311 | 18.0 | 30.3 | 0.37 | 9.2 | -40.2 ± 1.0 | -103.0 ± 1.7 | -138.8 ± 2.0 |
| Talca | Soils | GFNG1000R | Arid | -30.0545 | -71.0940 | 18.2 | 30.2 | 0.41 | 27.3 | -42.6 ± 0.9 | -160.9 ± 0.1 | -166.7 ± 3.7 |
| Talca | Soils | GFNG1000J | Arid | -30.0548 | -71.0894 | 12.9 | 30.2 | 0.43 | 28.5 | -42.6 ± 0.9 | -153.5 ± 2.4 | -159.5 ± 0.6 |



| Sampling Site | Sediment type | IGSN | Aridity Zone | Sampling Location Lat (°) | Sampling Location Long (°) | CPI | ACL | C29/(C29+C31) | Concentration Total n-alkanes ug g⁻¹ sediment dry weight | $\delta^2H$ precipitation [‰] | $\delta^2H$ $n$-C$_{29}$ [‰] | $\delta^2H$ $n$-C$_{31}$ [‰] |
|---|---|---|---|---|---|---|---|---|---|---|---|---|
| Cajon del Maipo | Soils | GFNG1000G | SemiArid | -33.5814 | -70.3586 | 14.7 | 29.9 | 0.59 | 67.5 | -69.7 ± 2.6 | -164.7 ± 0.6 | -162.3 ± 3.7 |
| SanAntonio-Maipo | Soils | GFNG1000K | SemiArid | -33.7736 | -71.5246 | 14.5 | 30.2 | 0.47 | 12.7 | -44.1 ± 0.6 | -182.7 ± 1.1 | -190.0 ± 6.5 |
| Rapel | Soils | GFNG1000L | SemiArid | -34.5693 | -71.6361 | 11.3 | 30.3 | 0.36 | 10.7 | -44.0 ± 0.8 | -170.0 ± 0.1 | -164.9 ± 2.9 |
| BioBio | Soils | GFNG1000E | Humid | -37.1126 | -72.9619 | 7.1 | 29.4 | 0.57 | 9.8 | -34.3 ± 1.4 | -178.7 ± 1.3 | -162.8 ± 2.4 |
| BioBio | Soils | GFNG1000M | Humid | -37.3473 | -72.8850 | 9.1 | 29.9 | 0.52 | 10.6 | -38.7 ± 1.4 | -164.2 ± 2.2 | -169.5 ± 1.9 |
| CalleCalle | Soils | GFNG1000N | Humid | -39.7885 | -72.9681 | 7.1 | 28.6 | 0.68 | 29.2 | -55.0 ± 0.4 | -160.8 ± 1.9 | -170.4 ± 6.2 |
| Bueno | Soils | GFNG1000P | Humid | -40.3326 | -73.4128 | 9.0 | 29.5 | 0.42 | 12.0 | -57.8 ± 0.8 | -163.8 ± 3.2 | -157.5 ± 1.4 |
| Maullin | Soils | GFNG1000H | Humid | -41.4740 | -73.2726 | 8.2 | 29.8 | 0.47 | 41.5 | -50.0 ± 1.5 | -173.9 ± 7.9 | -171.3 ± 2.6 |
| Maullin | Soils | GFNG1000Q | Humid | -41.4787 | -73.2756 | 11.0 | 29.7 | 0.51 | 93.4 | -50.0 ± 1.5 | -185.2 ± 0.7 | -179.2 ± 0.4 |
| GeoB7118-1_1-2cm | Marine | GEOB0071RX05V11 | | -25.9997 | -70.8092 | 2.3 | 30.8 | 0.36 | 0.3 | | -108 ± 4 | -105 ± 5 |
| GeoB7116-1_1-2cm | Marine | GEOB0071RXY2521 | | -26.0002 | -70.9998 | 6.6 | 30.3 | 0.41 | 0.1 | | -119 ± 3 | -115 ± 11 |
| GeoB7123-2_1-2cm | Marine | GEOB0071RX33521 | | -27.2900 | -71.0500 | 9.4 | 29.8 | 0.42 | 0.9 | | -151 ± 5 | -164 ± 14 |
| GeoB3377-1_1-2cm | Marine | GEOB0033RXS4V11 | | -27.4667 | -71.5250 | 5.9 | 29.3 | 0.52 | 0.5 | | -146 ± 13 | -153 ± 9 |
| GeoB3376-2_1-2cm | Marine | GEOB0033RXR4V11 | | -27.4667 | -71.3617 | 5.7 | 29.6 | 0.47 | 0.5 | | -145 ± 15 | -153 ± 11 |
| GeoB3374-1_1-2cm | Marine | GEOB0033RXQ4V11 | | -27.4733 | -71.1717 | 6.2 | 29.8 | 0.44 | 0.5 | | -145 ± 9 | -151 ± 9 |
| GeoB7127-1_1-2cm | Marine | GEOB0071RX15V11 | | -28.3837 | -71.4712 | 8.1 | 29.5 | 0.51 | 0.7 | | -152 ± 1 | -160 ± 3 |
| GeoB7129-1_1-2cm | Marine | GEOB0071RX25V11 | | -28.4168 | -71.3300 | 8.6 | 30.0 | 0.46 | 0.2 | | -156 ± 11 | -160 ± 2 |
| GeoB7130-1_1-2cm | Marine | GEOB0071RX83521 | | -28.4200 | -71.6130 | 8.3 | 29.8 | 0.46 | 0.4 | | -152 ± 5 | -158 ± 5 |
| GeoB7135-1_1-2cm | Marine | GEOB0071RX16V11 | | -29.6667 | -71.6758 | 7.7 | 29.6 | 0.48 | 0.7 | | -156 ± 1 | -162 ± 0 |
| GeoB7134-1_1-2cm | Marine | GEOB0071RXD3521 | | -29.7200 | -71.7700 | 5.4 | 30.2 | 0.46 | 0.4 | | -153 ± 4 | -159 ± 2 |
| GeoB7138_1-2cm | Marine | GEOB0071RXI3521 | | -30.1300 | -71.8700 | 9.1 | 30.2 | 0.46 | 1.1 | | -155 ± 5 | -161 ± 9 |
| GeoB7144-1_1-2cm | Marine | GEOB0071RXN3521 | | -31.1600 | -71.9700 | 8.8 | 30.1 | 0.45 | 1.3 | | -152 ± 0 | -155 ± 1 |
| GeoB3304-3_1-2cm | Marine | GEOB0033RX25521 | | -32.8900 | 72.1933 | 4.1 | 29.6 | 0.48 | 0.8 | | -154 ± 3 | -164 ± 2 |
| GeoB3303-1_1-2cm | Marine | GEOB0033RXX4521 | | -33.2067 | -72.2000 | 4.5 | 29.7 | 0.48 | 1.3 | | -158 ± 5 | -161 ± 0 |
| GeoB3311-2_1-2cm | Marine | GEOB0033RXL4V11 | | -33.6067 | -72.0467 | 4.1 | 29.4 | 0.49 | 1.9 | | -157 ± 4 | -167 ± 4 |
| GeoB7152-1_1-2cm | Marine | GEOB0071RX36V11 | | -33.8000 | -72.1102 | 6.5 | 30.2 | 0.49 | 1.5 | | -168 ± 0 | -172 ± 2 |
| GeoB3352-2_1-2cm | Marine | GEOB0033RXO4V11 | | -35.2167 | -73.3167 | 6.2 | 30.0 | 0.48 | 2.4 | | -166 ± 1 | -171 ± 3 |
| GeoB3355-4_1-2cm | Marine | GEOB0033RXP4V11 | | -35.2183 | -73.1167 | 7.2 | 30.1 | 0.47 | 2.1 | | -163 ± 1 | -167 ± 1 |
| GeoB7157-1_1-2cm | Marine | GEOB0071RX24521 | | -35.7800 | -73.5900 | 6.9 | 30.1 | 0.47 | 2.1 | | -164 ± 3 | -168 ± 1 |
| GeoB7160-4_1-2cm | Marine | GEOB0071RX66V11 | | -36.0385 | -73.0735 | 7.4 | 29.8 | 0.49 | 2.6 | | -168 ± 2 | -172 ± 0 |
| GeoB7167-4_1-2cm | Marine | GEOB0071RX74521 | | -36.4500 | -73.9100 | 3.7 | 30.2 | 0.47 | 2.9 | | -150 ± 3 | -153 ± 5 |
| GeoB7162-4_1-2cm | Marine | GEOB0071RX76V11 | | -36.5427 | -73.6672 | 6.1 | 29.9 | 0.48 | 3.5 | | -165 ± 0 | -170 ± 0 |
| GeoB7198-1_1-2cm | Marine | GEOB0071RXI4521 | | -38.1700 | -74.3900 | 4.8 | 29.8 | 0.48 | 2.0 | | -151 ± 3 | -150 ± 3 |
| GeoB7209-2_1-2cm | Marine | GEOB0071RXB6V11 | | -38.9913 | -74.1642 | 5.1 | 29.6 | 0.47 | 2.1 | | -161 ± 6 | -163 ± 4 |
| GeoB7207-1_1-2cm | Marine | GEOB0071RXN4521 | | -39.0700 | -74.3700 | 4.0 | 29.9 | 0.47 | 2.5 | | -166 ± 4 | -164 ± 5 |
| GeoB7212-1_1-2cm | Marine | GEOB0071RXS4521 | | -39.7000 | -74.3800 | 8.5 | 29.5 | 0.48 | 1.6 | | -154 ± 3 | -155 ± 3 |
| GeoB7214-1_1-2cm | Marine | GEOB0071RXD6V11 | | -39.8750 | -74.1672 | 5.3 | 29.6 | 0.46 | 0.8 | | -156 ± 2 | -154 ± 7 |
| GeoB7194-1_1-2cm | Marine | GEOB0071RXA6V11 | | -41.4175 | -74.4337 | 5.9 | 29.7 | 0.47 | 2.2 | | -164 ± 4 | -161 ± 3 |

### 3.1.2 Concentration and distribution of n-alkanes in marine sediments

All 29 marine sediments analyzed presented odd over even chain length predominance with a CPI between 2.3 and 9.4 (Table 1). The *n*-alkane concentration was lowest (0.1 µg g⁻¹ dw) in sediments adjacent to the hyperarid zone and highest (3.5 µg g⁻¹ dw) in sediments adjacent to the humid zone (Table 1). In the marine sediments, chain length distribution of the *n*-alkanes varied with an ACL ranging from 29.3 to 30.8 (Table 1). The highest ACL values were found in sediments adjacent to the hyperarid zone (30.8), but no clear trend with latitude was apparent in the ACL values of the marine sediments. The most abundant *n*-alkane homologue in the studied marine surface sediments was *n*-C$_{31}$, except for two samples in the hyperarid zone, where *n*-C$_{29}$ concentration was higher (Table 1, Table S4).



### 3.2 $\delta^2H_{wax}$ values of river, soil, and marine sediments

### 3.2.1 $\delta^2H_{wax}$ values in river sediments and soils

The $\delta^2H_{wax}$ values of the $n$-$C_{29}$ and $n$-$C_{31}$ homologues varied from -90‰ to -185‰, and from -114‰ to -190‰ respectively (Table 1). The mean $\delta^2H_{wax}$ values from $n$-$C_{29}$ in the hyperarid and arid zones were found to be 8‰ less negative than the mean values from $n$-$C_{31}$, while in the humid zone, the mean $\delta^2H_{wax}$ values from $n$-$C_{29}$ were 1‰ more negative compared to those from $n$-$C_{31}$.

**Table 2:** Statistical parameters of the linear regressions between $\delta^2H_{pre}$ and $\delta^2H_{wax}$. The regressions obtained are of the type Y=Intercept +
Slope*X df indicates the number of degrees of freedom from the fitted linear model and F is the F-statistic number.

| X | Y | Dataset | Sediment Type | Slope | Intercept | df | R2 | F | p value |
|---|---|---|---|---|---|---|---|---|---|
| $\delta^2H$ precipitation | $\delta^2H$ $n$-$C_{29}$ | Chilean | Soils | 0.82 ± 0.82 | -126 ± 38 | 10 | 0.09 | 1 | 3E-01 |
|  |  | Chilean | Rivers | 1.40 ± 0.33 | -79 ± 18 | 24 | 0.43 | 18 | 3E-04 |
|  |  | Chilean | Soils+Rivers | 0.76 ± 0.31 | -118 ± 17 | 36 | 0.14 | 6 | 2E-02 |
|  |  | Global | Soils+Lakes | 0.79 ± 0.02 | -124 ± 1 | 748 | 0.71 | 1835 | 2E-203 |
| $\delta^2H$ precipitation | $\delta^2H$ $n$-$C_{31}$ | Chilean | Soils | 0.41 ± 0.49 | -147 ± 22 | 10 | 0.07 | 1 | 4E-01 |
|  |  | Chilean | Rivers | 1.05 ± 0.21 | -101 ± 12 | 24 | 0.52 | 26 | 3E-05 |
|  |  | Chilean | Soils+Rivers | 0.51 ± 0.21 | -135 ± 11 | 36 | 0.14 | 6 | 2E-02 |
|  |  | Global | Soils+Lakes | 0.76 ± 0.02 | -130 ± 1 | 661 | 0.73 | 1753 | 5E-188 |

We examined the relationship between the measured $\delta^2H_{wax}$ and $\delta^2H_{pre}$ derived from the OIPC (Fig. 2A, 2C). We found that $n$-$C_{29}$ and $n$-$C_{31}$ $\delta^2H_{wax}$ values from our Chilean river sediments show positive correlation with $\delta^2H_{pre}$ ($\delta^2H$$n$-$C_{29}$: $R^2 = 0.43$, p <0.001; $n$-$C_{31}$: $R^2 = 0.52$, p <0.001) (Table 2), while $\delta^2H_{wax}$ from Chilean soils show no significant correlation with $\delta^2H_{pre}$ ($n$-
$C_{29}$: $R^2 = 0.06$, p = 0.41; $n$-$C_{31}$: $R^2 = 0.04$, p = 0.56) (Table 2). Analyzing rivers and soils $\delta^2H_{wax}$ together vs $\delta^2H_{pre}$ we found a low value, yet still statistically significant correlation ($n$-$C_{29}$: $R^2 = 0.14$, p = 0.02; $n$-$C_{31}$: $R^2 = 0.14$, p = 0.02) (Table 2).

For the global compiled dataset, we found a strong significant correlation between $\delta^2H_{wax}$ and $\delta^2H_{pre}$ ($n$-$C_{29}$: $R^2 = 0.71$, p <0.001; $n$-$C_{31}$: $R^2 = 0.73$, p <0.001) (Table 2, Fig. 2A, 2C). Chilean $\delta^2H_{wax}$ values from the humid, semiarid and arid zones generally follow the linear relationship between $\delta^2H_{wax}$ and $\delta^2H_{pre}$ established by the global dataset (Fig. 2A, 2C), while $\delta^2H_{wax}$
values from the hyperarid zone exhibit a noticeable departure from the global linear relationship between $\delta^2H_{wax}$ and $\delta^2H_{pre}$ (Fig. 2.A, 2C).

**Table 3.** Results of Kruskal-Wallis tests performed to evaluate the difference in residual values across different aridity zones.

|  | Arid-Semiarid-Humid | | | Hyperarid-Arid-Semiarid-Humid | | |
|---|---|---|---|---|---|---|
|  | p-value | chi-squared | df | p-value | chi-squared | df |
| **n-C$_{29}$** | 0.124 | 4.184 | 2 | 2.17E-04 | 19.484 | 3 |
| **n-C$_{31}$** | 0.545 | 1.216 | 2 | 1.87E-03 | 14.941 | 3 |





From our measured samples, we used $\delta^2H_{wax}$ values from river and soil samples to calculate residuals, following the methodology described in section 2.6.1. $\delta^2H_{wax}$ values residuals, from $n$-$C_{29}$ and $n$-$C_{31}$, were highest in samples from the hyperarid zone (Fig. 2.B, 2.D and Table S5) and decreased with increasing humidity. $n$-$C_{29}$ mean residuals values ranged from 32‰ in the hyperarid zone to -3 ‰ in the humid zone, while $n$-$C_{31}$ residuals varied from 23‰ in the hyperarid zone to 4 ‰ in the humid zone. Residuals from $n$-$C_{29}$ and $n$-$C_{31}$ for the humid, semiarid, and arid zones showed no significant differences

(Kruskal-Wallis test, section 2.6.2; $n$-$C_{29}$: $p = 0.125$; $n$-$C_{31}$: $p = 0.545$; Table 3). When residuals from the hyperarid zone were analyzed together with the residuals from the other zones, the Kruskal-Wallis test indicated the existence of a significant difference among them ($n$-$C_{29}$: $p < 0.001$; $n$-$C_{31}$: $p < 0.002$; Table 3).

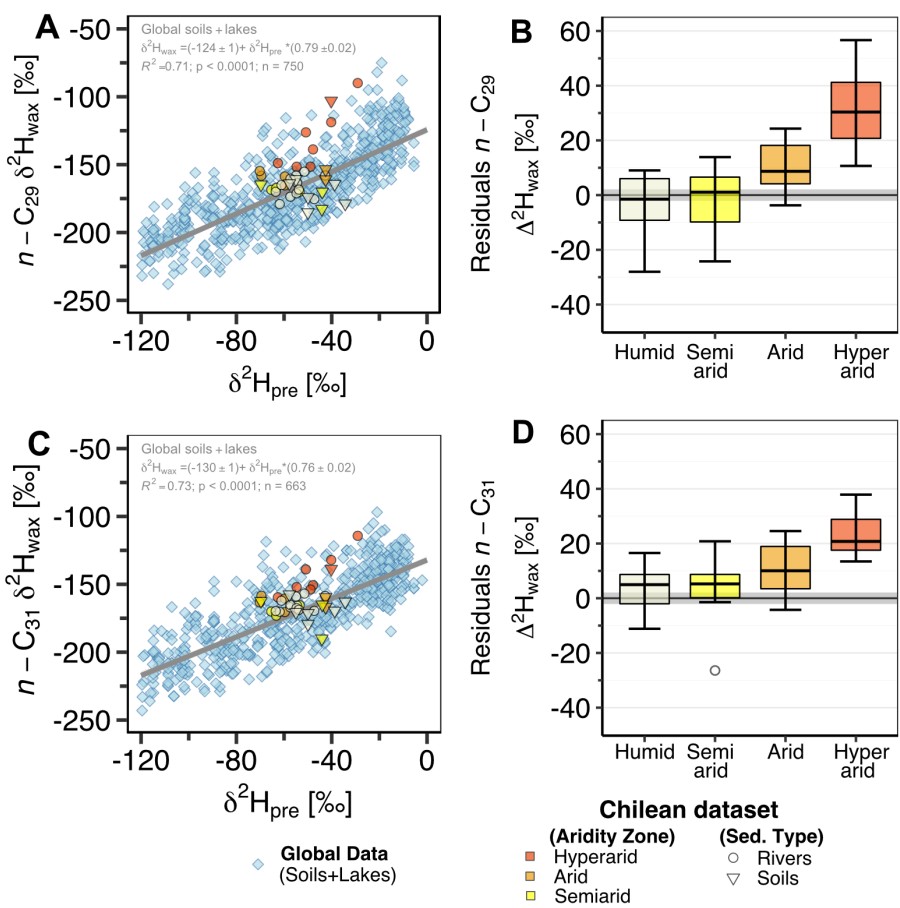

**Figure 2:** (**A+C**) $\delta^2H_{wax}$ values from $n$-$C_{29}$ and $n$-$C_{31}$ homologues vs. $\delta^2H_{pre}$. Inverted triangles represent Chilean soil samples, circles represent Chilean river sediment samples. The blue diamonds represent $\delta^2H_{wax}$ from a global compilation of previously published data from soils and lake surface sediments (Table S1). $\delta^2H_{pre}$ data for both the Chilean locations and the global compilation dataset are derived from OIPC (Bowen and Revenaugh, 2003). The grey line in the plot illustrates the linear relationship between $\delta^2H_{wax}$ and $\delta^2H_{pre}$ for the global dataset, as indicated by the equation and regression parameters annotated within the plot. (**B+D**) Boxplots of the calculated residuals from the Chilean sediments (soils + river sediments) with respect to the global regression. The aridity zone classification follows the classification proposed by UNEP (1997).




### 3.2.2 $\delta^2H_{wax}$ values in marine sediments

$\delta^2H_{wax}$ values of $n$-C$_{29}$ and $n$-C$_{31}$ in marine sediments varied from -108‰ to -168‰, and from -105‰ to -172‰, respectively (Table 1). For both $n$-C$_{29}$ and $n$-C$_{31}$, $\delta^2H_{wax}$ values were less negative in marine sediments adjacent to the continental hyperarid zone and more negative in sediments adjacent to the continental humid zone. $\delta^2H_{wax}$ values from $n$-C$_{29}$ were on average 4‰

less negative compared to values from $n$-C$_{31}$ along the gradient (Table 1).

### 3.3 Apparent hydrogen isotope fractionation along the Chilean gradient

Apparent fractionation values ($\varepsilon_{wax/pre}$) varied from -63‰ to -150‰, and from -88‰ to -153‰ for the $n$-C$_{29}$ and $n$-C$_{31}$ homologues, respectively (Table S6). Both soils and river sediments exhibited less negative $\varepsilon_{wax/pre}$ values in the hyperarid zone ($\varepsilon_{wax/pre}$ $n$-C$_{29}$ = -63‰; $\varepsilon_{wax/pre}$ $n$-C$_{31}$ = -88‰). In general, $\varepsilon_{wax/pre}$ values were increasingly negative with increasing

humidity.

We found significant non-linear relationships between $\varepsilon_{wax/pre}$ and water-climate parameters, i.e., actual evapotranspiration, mean annual precipitation, soil moisture, relative humidity, and aridity index (Table S7). Actual evapotranspiration showed the strongest correlation with $\varepsilon_{wax/pre}$ (river sediments $n$-C$_{29}$: $R^2 = 0.72$, p <0.0001; river sediments $n$-C$_{31}$: $R^2 = 0.61$, p <0.0001) (Fig. 3, Table S7). Parameters associated solely with temperature (i.e. mean annual temperature, mean annual max daily

temperature) showed no correlation with $\varepsilon_{wax/pre}$ (Table S7). The $\varepsilon_{wax/pre}$ values obtained from river samples were more strongly correlated with the climatic parameters than the $\varepsilon_{wax/pre}$ values from soils (Table S7).

## 4 Discussion

### 4.1 Causes of $\delta^2H_{wax}$ variability in sediments and soils along the Chilean aridity gradient

### 4.1.1 Assessment of the OIPC $\delta^2H_{pre}$ values along the aridity gradient

Analyzing the drivers of $\delta^2H_{wax}$ values requires robust data on $\delta^2H_{pre}$. Since no long-term time series are available for the exact locations of our sampling sites, we rely on interpolated data generated by the OIPC (see above). Thus, a first factor to consider in our interpretations is the accuracy of the $\delta^2H_{pre}$ values derived from the OIPC. The OIPC provides long-term estimates of modern (post 1950) mean $\delta^2H_{pre}$, at monthly and annual grids, which may differ from the $\delta^2H_{pre}$ measured at a particular site on a short-term basis. The longer-term estimates from the OIPC are advantageous for our study, as soils and river sediments

can integrate leaf waxes over spans of several decades to centuries (Huang et al., 1996; Douglas et al., 2014; Vonk et al., 2019). Consequently, we expect $\delta^2H_{pre}$ values from OIPC are the more relevant predictors for our $\delta^2H_{wax}$ values. The OIPC model is





based on data from GNIP stations (Bowen and Revenaugh, 2003), of which nine fall within continental Chile and can be used as a validation dataset. They range from the humid zone to the border between the arid and hyperarid zones and include a total of 923 monthly precipitation datapoints between 1964 and 2017 (Table S3). We find a significant linear relationship ($R^2 =$

0.53, p < 0.0001) between the $\delta^2 H_{pre}$ predicted by OIPC and measured $\delta^2 H_{pre}$ (Fig. S1.A). Grouping by month and taking the long-term average of the measured $\delta^2 H_{pre}$ values, we found that the correlation is stronger ($R^2 = 0.80$, p < 0.0001) (Fig. S1.B). Although there are uncertainties associated with the predicted OIPC $\delta^2 H_{pre}$ values, particularly in regions with sparse GNIP station coverage, the results presented above validate the use of OIPC $\delta^2 H_{pre}$ values across the studied aridity gradient in Chile.

### 4.1.2 $\delta^2 H_{pre}$ values control $\delta^2 H_{wax}$ values in the humid to arid zone

In the humid and semiarid zones of Chile $\delta^2 H_{pre}$ values are the main factor determining $\delta^2 H_{wax}$ values. Our results show that $\delta^2 H_{wax}$ values from soils and river sediments in the humid and semiarid zones of Chile generally follow the global linear relationship between $\delta^2 H_{wax}$ and $\delta^2 H_{pre}$. This is confirmed through the residuals analysis (Fig. 2.B, 2.D) and is in accordance with the findings of previous studies in humid and semiarid regions (Feakins et al., 2016; Hou et al., 2008; Sachse et al., 2006; Tipple and Pagani, 2013; Tuthorn et al., 2015; Häggi et al., 2016; Bertassoli et al., 2022). Therefore, we suggest that $\delta^2 H_{wax}$ is

primarily controlled by $\delta^2 H_{pre}$ in the humid and semiarid zones of Chile.

In the arid zone of Chile, $\delta^2 H_{wax}$ values show a slight deviation from the global regression between $\delta^2 H_{wax}$ and $\delta^2 H_{pre}$ (Fig. 2.B, 2.C). However, this deviation is not statistically significant when compared to the deviation of $\delta^2 H_{wax}$ values from the humid and semiarid zones of Chile (Table 3). As a result, the net or apparent fractionation between water source and lipid biomarker ($\epsilon_{wax/pre}$) is not significantly different in the arid zone when compared to humid and semiarid zones. Although previous studies

have found that $\delta^2 H_{wax}$ values in arid zones are consistently less negative due to changes in $\epsilon_{wax/pre}$ (Douglas et al., 2012; Feakins and Sessions, 2010; Polissar and Freeman, 2010), we can not resolve a statistically significant difference between the arid zone, the humid zone and semiarid zone (Fig. 2.B, 2.C, Table 3). This might be because earlier studies including arid zones focused on lakes draining small areas or exclusively analyzed soil and plant samples (Douglas et al., 2012; Feakins and Sessions, 2010; Polissar and Freeman, 2010; Schwab et al., 2015). Our study analyzed only two soil samples from the arid

zone, which may not be a complete representation of the whole $\delta^2 H_{wax}$ variability. It is important to note that $\delta^2 H_{wax}$ values are significantly more variable in soils than in river sediments in all climate zones (as discussed in section 4.4). Therefore, further sampling at smaller subcatchments or at the soil scale is necessary to confidently test if $\delta^2 H_{wax}$ largely reflects $\delta^2 H_{pre}$ in the arid zone.



### 4.1.3 Controls on $\delta^2H_{wax}$ variability in the hyperarid zone

$\delta^2H_{wax}$ values in the hyperarid zone of Chile are influenced by additional factors beyond $\delta^2H_{pre}$ values as suggested by the Kruskal-Wallis test results (Table 3) and the deviation of $\delta^2H_{wax}$ values from the global linear relationship between $\delta^2H_{wax}$ and $\delta^2H_{pre}$ (Fig. 2.A, 2.C). Along aridity gradients, climatic parameters such as evapotranspiration, relative humidity, and aridity itself have been identified as key factors that exert control over $\varepsilon_{wax/pre}$ values (Douglas et al., 2012; Polissar and Freeman, 2010; Schwab et al., 2015; Herrmann et al., 2017; Li et al., 2019). The mechanism behind this control is rooted in the impact

that these climatic parameters have on soil water enrichment and leaf water enrichment (Gat, 1996; Smith and Freeman, 2006; Kahmen et al., 2013a). To determine the factors controlling $\varepsilon_{wax/pre}$ along the Chilean aridity gradient, we conducted a regression analysis as outlined in section 2.6.3. This analysis examined $\varepsilon_{wax/pre}$ against three broad categories of climatic factors: temperature, water content, and water fluxes. Temperature was evaluated based on maximum daily temperature (MaxT and mean annual temperature (MAT). Water content in the soil and atmosphere was evaluated through relative humidity (RH), soil

moisture (SM), and aridity index (AIdx). Water fluxes were analyzed through actual/net evapotranspiration (AET) and mean annual precipitation (MAP). To further validate our findings, we used data from four previously published aridity gradients to determine if the observed trends are characteristic of strong aridity gradients globally, or simply a feature of the Chilean dataset. We selected four regions with the most pronounced aridity gradients in our compilation (Table S2), comprising data from soils from Argentina (Tuthorn et al., 2015), China (Rao et al., 2009; Li et al., 2019; Lu et al., 2020), Israel (Goldsmith et al., 2019),

and South Africa (Herrmann et al., 2017; Strobel et al., 2020). To maintain the rigor of our analysis, we adopted a cautious approach and excluded publications that contained marked altitude gradients within the four selected regions, thereby eliminating the potentially confounding effect of elevation.

Our regression analysis indicates that no significant relationship exists between MAT or MaxT and $\varepsilon_{wax/pre}$ (Table S7), which is consistent with mechanistic models of leaf water enrichment, which demonstrate a low sensitivity to variations in

temperature (Farquhar and Gan, 2003; Kahmen et al., 2011; Rach et al., 2017). Furthermore, field studies in climatic transects along South Africa from Strobel et al. (2020) also found no significant correlation between temperature and $\varepsilon_{wax/pre}$. Similarly, along an aridity gradient in Argentina, Tuthorn et al. (2015) showed that relative humidity exerts a greater control than temperature on the magnitude of $^2H$ enrichment in leaf waxes. Thus, the narrow MAT range of 8°C to 17°C along our study area is expected to have minimal impact on leaf water enrichment and consequently on $\varepsilon_{wax/pre}$.

The significant correlations that we found between $\varepsilon_{wax/pre}$ and variables linked to water content and water fluxes (Table S7) highlight the influence of hydrological conditions on $\varepsilon_{wax/pre}$ values. Results from a principal component analysis (Fig. S2) demonstrate that hydrological variables are correlated, contributing to 69% of the total variance observed. This confirms the synchronized variation of these parameters, due to the hydrological cycle's water balance, and to some degree, their autocorrelation (Douglas et al., 2012). Similarly, all hydrological variables are strongly correlated with $\varepsilon_{wax/pre}$ in our study,



but AET exhibits the highest correlation (Fig. 3, Table S7). Hence, we suggest that in the hyperarid zone of Chile, $\delta^2H_{wax}$ values are strongly controlled by evapotranspirative processes, in addition to $\delta^2H_{pre}$ values.

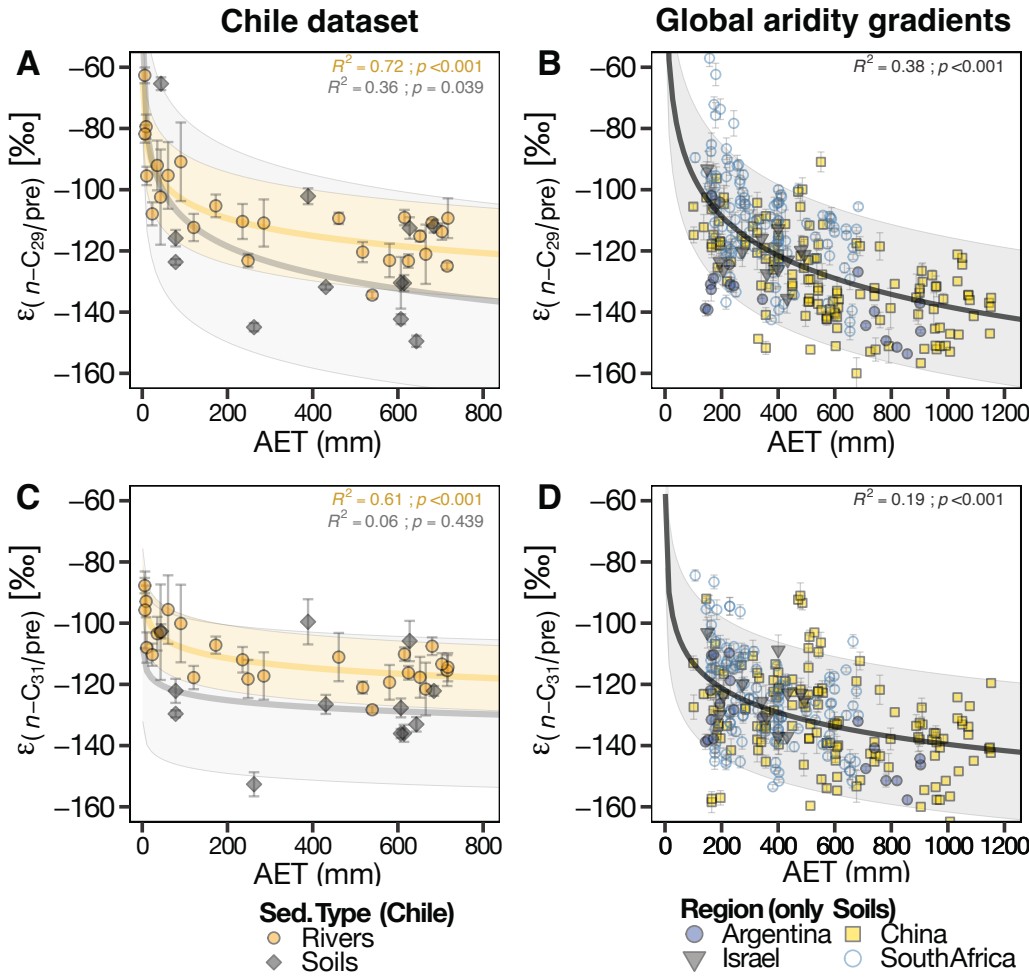

**Figure 3:** $\varepsilon_{wax/pre}$ vs. AET (actual evapotranspiration) (**A+C**) Chilean soils and river sediments. Grey shaded area represents 95% CI of the model fitted to the soil data, yellow shaded area represents 95% CI of the model fitted to the river sediments. (**B+D**) Soils from aridity gradients of Argentina (Tuthorn et al., 2015), China (Rao et al., 2009; Li et al., 2019; Lu et al., 2020), Israel (Goldsmith et al., 2019), and South Africa (Herrmann et al., 2017; Strobel et al., 2020) grey shaded area represents 95% CI of the model fitted to the full dataset.

### 4.1.4 Soil and leaf water enrichment modelling in the hyperarid zone

We modeled the $^2H$ enrichment in soil and leaf water of river samples in the hyperarid zone using Eq. (9) and Eq. (20). By comparing the model results with the enrichment measured in our samples, while holding vegetation parameters constant, we can isolate the contribution of climatic factors to the changes in $\varepsilon_{wax/pre}$ along the gradient. Eq. (9) and Eq. (20) produce a net soil ($\Delta^2H_{SW}$) and net leaf water enrichment ($\Delta^2H_{LWP}$) values, respectively, for each sample. To make the modeled $\Delta^2H_{SW}$ and




$\Delta^2H_{LWP}$ values comparable with empirical $\varepsilon_{wax/pre}$ values, we standardize the values of each catchment relative to the values obtained for the southernmost catchment of the hyperarid zone (Los Choros) (Table S8). From the standardization of the $\varepsilon_{wax/pre}$ values we obtain an empirical relative net $^2H$ enrichment for each catchment ($\Delta^2H_{REm}$). The standardization of $\Delta^2H_{SW}$ and

$\Delta^2H_{LWP}$ yields a relative $\Delta^2H_{SW}$ and a relative $\Delta^2H_{LWP}$, the sum of which yields a modeled relative $^2H$ enrichment for each catchment ($\Delta^2H_{RMd}$) (Table S8). The progressive aridification trend towards the north offers an opportunity to evaluate the consistency between $\Delta^2H_{REm}$ and $\Delta^2H_{RMd}$ in response to aridity. Additionally, this approach eliminates the uncertainty surrounding the absolute value of biosynthetic fractionation, which would otherwise be a requirement for comparison of the $\Delta^2H_{SW}$ and $\Delta^2H_{LWP}$ against $\varepsilon_{wax/pre}$.

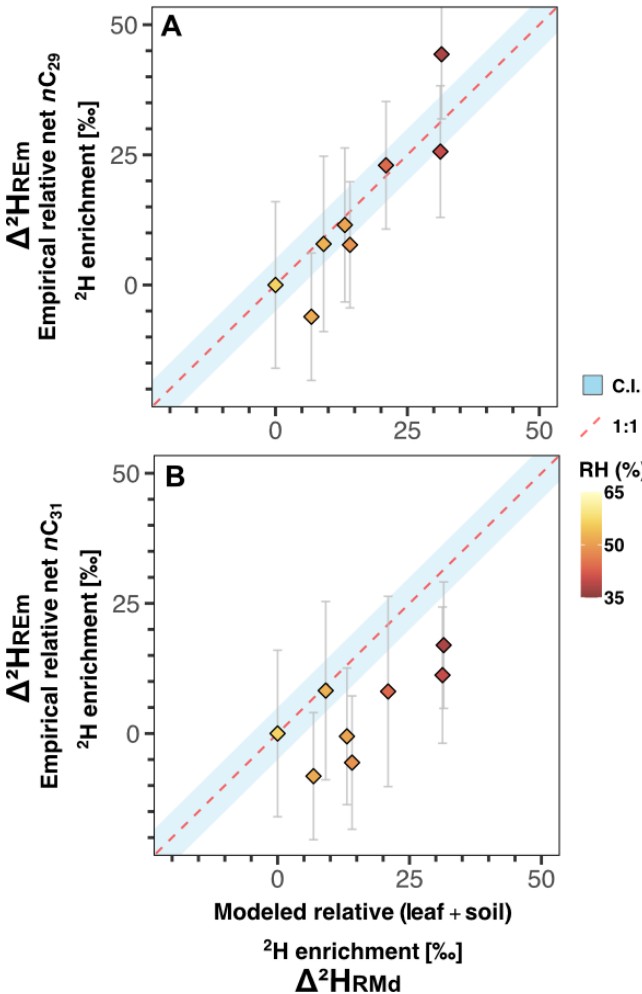

**Figure 4:** $\Delta^2H_{REm}$ vs $\Delta^2H_{RMd}$ for the catchments of the hyperarid zone. The red dashed line represents a 1:1 line, an analytical uncertainty range of 5‰ is used to represent the confidence interval (blue shaded area). Markers are color-coded by relative humidity (RH). **A.** Results obtained for the *n*-C$_{29}$ homologue. **B.** Results obtained for the *n*-C$_{31}$ homologue.



Our modeling approach was able to reproduce well the empirical enrichment measured in $n$-$C_{29}$, but the enrichment measured
in $n$-$C_{31}$ was overestimated (Fig. 4.A, 4.B). Despite the high uncertainties propagated into the empirical relative enrichment
values, the agreement between modeled and empirical values for $n$-$C_{29}$ supports the hypothesis that evapotranspiration
processes play a significant role in the fractionation of hydrogen isotopes in the hyperarid zone of Chile. Although for $n$-$C_{31}$
most of the empirical enrichment is reproduced within uncertainty, there is a clear overestimation of the enrichment by the
model (Fig. 4.B). The model overestimation for $n$-$C_{31}$ indicates that changes in climatic conditions alone do not account for
all the variability in the magnitude of the enrichment. Some residual variability can be caused by vegetation effects that
generate a differential enrichment in $n$-$C_{29}$ and $n$-$C_{31}$ among different plant types, this is further discussed in the next section
(4.2).

### 4.2 Aridity highlights $\delta^2 H_{wax}$ differences among n-$C_{29}$ and n-$C_{31}$ n-alkanes

In arid sites dominated by herbaceous vegetation, $n$-$C_{29}$ consistently exhibited less negative $\delta^2 H_{wax}$ values compared to $n$-$C_{31}$
(Fig. 5, Table S6). This suggests a differential response of the homologues to evapotranspirative processes, which can be
attributed to the plant sources from which they originate. Woody plants like trees and shrubs generally yield higher
concentrations of $n$-$C_{29}$ than $n$-$C_{31}$, and herbaceous plants like grasses and forbs tend to produce higher concentrations of $n$-
$C_{31}$ than $n$-$C_{29}$ (Kuhn et al., 2010; Zech et al., 2010; Duan and He, 2011; Howard et al., 2018; Bliedtner et al., 2018).
Herbaceous plants possess different strategies to cope with aridity, including the development of smaller leaves and the use of
CAM photosynthesis. They also tend to have deeper rooting depths, which gives them access to water sources less enriched in
$^2H$ than surficial soil water (Ehleringer et al., 1991; Gibson, 1998; Gibbens and Lenz, 2001; Herrera, 2009; Feakins and
Sessions, 2010; Kirschner et al., 2021). Additionally, greenhouse experiments have demonstrated that grasses use a mixture
of enriched leaf water and less enriched soil water for biosynthesis (Kahmen et al., 2013a). Furthermore, plant physiology,
biochemistry, or even the timing of leaf flush can have a significant impact on $\delta^2 H_{wax}$ (Bi et al., 2005; Liu et al., 2006; Smith
and Freeman, 2006; Hou et al., 2007; Liu and Yang, 2008; Sachse et al., 2012; Tipple et al., 2013). Thus, we interpret the
differences in $\delta^2 H_{wax}$ values of $n$-$C_{31}$ and $n$-$C_{29}$ homologues as the manifestation of distinct physiological and biochemical
characteristics among different plant growth forms. Notably, the differences in $\delta^2 H_{wax}$ values among homologues become
significantly pronounced only under conditions of high aridity.

Our findings offer empirical evidence that disparities in $\delta^2 H_{wax}$ values between $n$-$C_{31}$ and $n$-$C_{29}$ homologues can be utilized as
a marker of specific vegetation and aridity conditions, such as the predominance of herbaceous plants under high aridity. Some
studies noted and briefly discussed the divergence in $\delta^2 H_{wax}$ values from the $n$-$C_{29}$ and $n$-$C_{31}$ homologues (Garcin et al., 2012;
Wang et al., 2013; Chen et al., 2022), but to our knowledge this has not been further analyzed. The results of this study suggest
that differences between the homologues contain valuable information that could potentially be exploited as a proxy to indicate
both aridity and the presence of herbaceous plants, but further research is needed to validate its application.





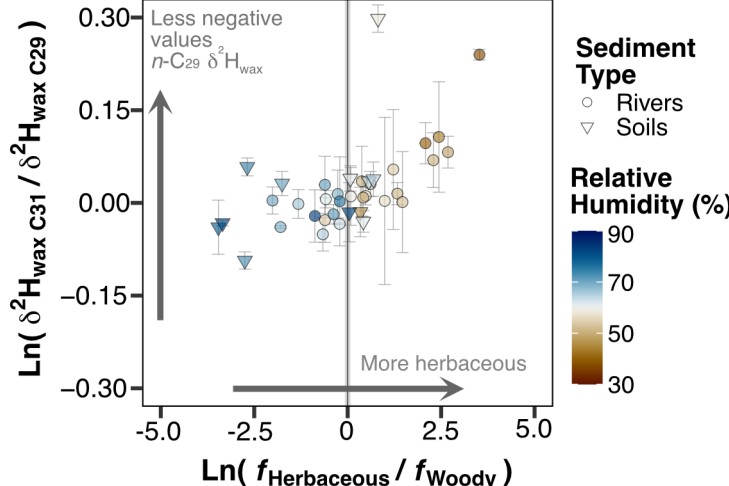

**Figure 5:** Plot of log ratio between $n$-$C_{31}$ $\delta^2H_{wax}$ and $n$-$C_{29}$ $\delta^2H_{wax}$ versus the log ratio between the fraction of herbaceous plants and fraction of woody plants. The fractions of herbaceous and woody vegetation were obtained following the described in the methods section 2.5.3. The vegetation cover is derived at the scale of the catchment area for river sediment samples and at the maximum resolution of the pixel (100 m x100 m) for soil samples. Vegetation cover fraction data is from Buchhorn et al. (2020).

### 4.3 Non-linear relationship between $\varepsilon_{wax/pre}$ and hydrological factors

The results of the regression analyses shown in Table S7 revealed that $\varepsilon_{wax/pre}$ is non-linearly correlated with all the studied hydrological parameters. The $\varepsilon_{wax/pre}$ values in Chilean soils exhibit non-linear correlations with hydrological factors. However, for Chilean soils, the only significant correlations between $\varepsilon_{wax/pre}$ and climatic variables are with SM and AET, these correlations are observed only for the $n$-$C_{29}$ homologue. This likely reflects the low sample density of soils in our study, particularly in the arid and hyperarid regions. For river samples as well as for the four selected aridity gradients the results show that all regressions against hydrological factors are significant and non-linear (Table S7). These results indicate that the non-linear relationship between $\varepsilon_{wax/pre}$ and hydrological factors can be found on strong aridity gradients globally and is not merely an artifact of our data.

The non-linear relationship between enrichment and hydrological conditions can be modeled combining the leaf and soil water enrichment models exposed in section 2.6.4 of the methods. By parametrizing RH and MAP in terms of AET, Figure 6 shows that the non-linear behavior can be reproduced when the model includes a component of soil evaporation ($f$e > 0). These modelling results support our findings and strengthen the idea that under high aridity, the relationship between $\varepsilon_{wax/pre}$ and hydrological conditions is non-linear. Previous studies showed that $\varepsilon_{wax/pre}$ is controlled by hydrological conditions (Hou et al., 2018; Smith and Freeman, 2006; Feakins and Sessions, 2010; Sachse et al., 2012; Douglas et al., 2012; Herrmann et al., 2017; Li et al., 2019; Lu et al., 2020), but most argue that the relationship between $\varepsilon_{wax/pre}$ and the hydrological parameters is linear (Feakins and Sessions, 2010; Vogts et al., 2016; Herrmann et al., 2017; Li et al., 2019). By extensive including and analysing



samples from hyperarid regions, we demonstrate that $\varepsilon_{wax/pre}$ behaves non-linearly along strong aridity gradients, as is also predicted by enrichment models incorporating both soil evaporation and leaf water transpiration (Fig. 6).

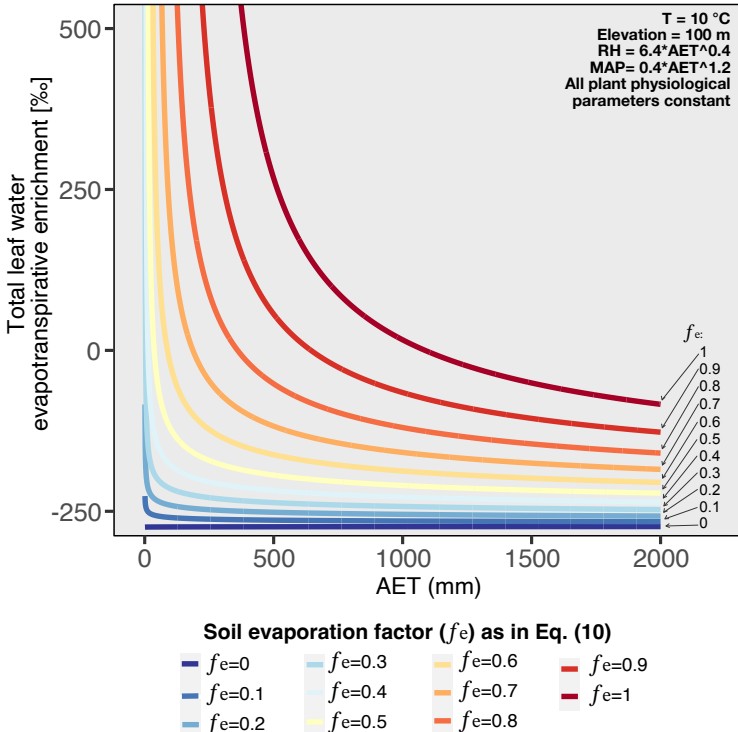

**Figure 6:** Modeled total leaf water evapotranspirative enrichment under varying AET, considering different soil evaporation factors ($f_e$). The $f_e$ values are used to derive the soil evaporated water from AET as expressed in Eq. (10). The total evapotranspirative leaf water enrichment is obtained summing the results of Eq. (9) and Eq. (20). To obtain the results for this figure RH and MAP were parametrized as a function of AET (see Fig. S3 and Fig. S4), as such all the variation in the evapotranspirative enrichment is only driven by variation of AET.

**4.4 $\delta^2H_{wax}$ differences between soils, riverine and marine sediments**

Along the Chilean aridity gradient, soil $\delta^2H_{wax}$ values were more variable than river $\delta^2H_{wax}$ values. In addition, soil $\delta^2H_{wax}$ values showed weaker correlations with all climatic variables than river $\delta^2H_{wax}$ values. This suggests a larger effect of spatial averaging of the $\delta^2H_{wax}$ values for the river sediment samples, as they average both vegetation and climatic variability to a greater extent than soil samples, as they integrate over larger regions. In support of this, Goldsmith et al. (2019) showed a significant reduction of variability via the early incorporation of the leaf waxes into soils, finding that variability of $\varepsilon_{wax/pre}$ from soils is significantly lower than variability of $\varepsilon_{wax/pre}$ from plants. In a similar manner, lake sediments exhibit lower $\varepsilon_{wax/pre}$ variability than both plants (Sachse et al., 2006) and soils (Douglas et al., 2012) in their source areas.





Overall, the ranges of $\delta^2H_{wax}$ values from soils, rivers and marine sediments overlap (Fig. 7), but there are two notable outliers in marine sediments of the hyperarid zone (Fig. 7). These samples were taken adjacent to the Pan de Azúcar catchment at 25.9

500 °S ($\delta^2H_{waxC29}$ = -108 ‰) and at 26.0 °S ($\delta^2H_{waxC29}$ = -119 ‰) and have similar $\delta^2H_{wax}$ values as continental samples from locations further south. The Quebrada Puerto Flamenco at 26.5 °S ($\delta^2H_{waxC29}$ = -119 ‰) and Quebrada Salitrosa at 26.1 °S ($\delta^2H_{waxC29}$ = -90 ‰), which are catchments with predominantly herbaceous vegetation that span small areas close to the coastline. This suggests that these catchments could be a source of sediments to the identified outliers.

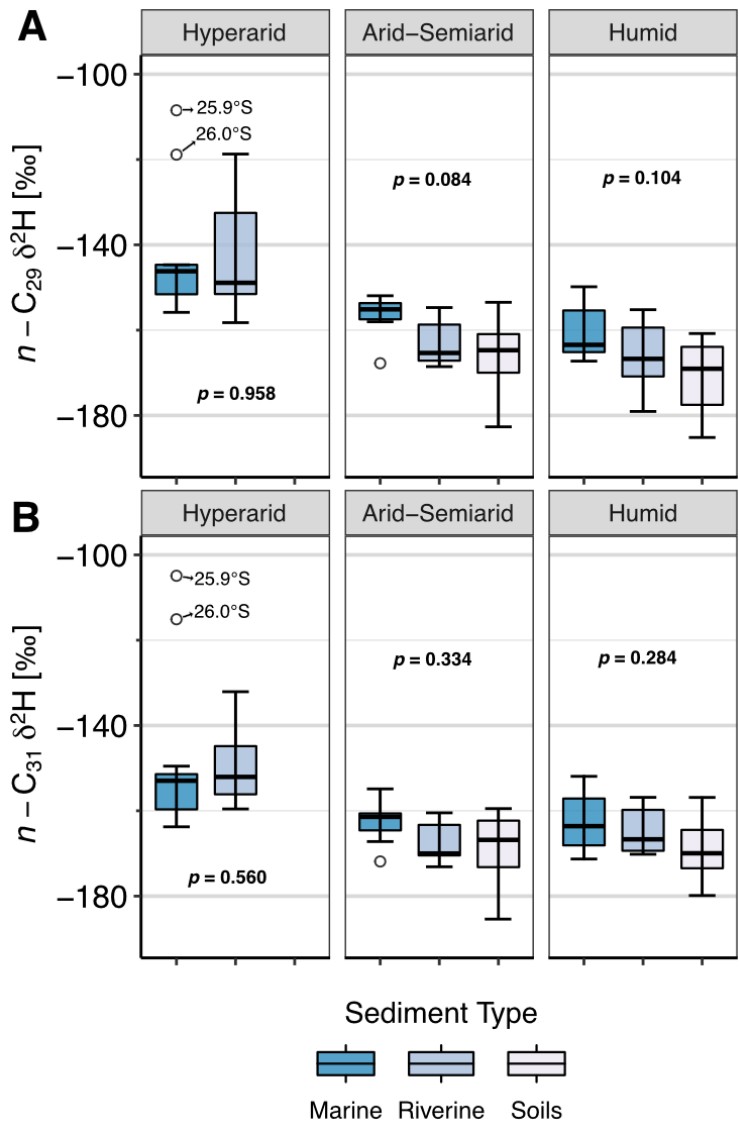

**Figure 7:** Boxplots of $\delta^2H_{wax}$ in soils, rivers and marine surface sediments, among the different climate zones. (A) Data for the *n*-C$_{29}$
homologue. (B) Data for the *n*-C$_{31}$ homologue. Arid and semiarid zones were grouped to overcome sample size limitations. In the hyperarid zone only one soil sample was taken, so no box is shown.





We conducted a Kruskal-Wallis test (Section 2.6.2) to assess the presence of significant differences in $\delta^2H_{wax}$ values among the soils, rivers and marine sediments from the different aridity zones. Together with the two marine outliers (above), nested catchments (Quebrada Salitrosa, Quebrada La Campana, and Estero Los Gringos) were excluded from the analysis, as the major catchments that include them would be expected to contribute the $\delta^2H_{wax}$ values transported to the marine sediments.

We find no statistical differences between mean $\delta^2H_{wax}$ values of the sediment types within each climate zone (Fig. 7, Table 4). This suggests that climatic signals recorded by $\delta^2H_{wax}$ are effectively transported along the sedimentary systems. In general, marine sediments show the smallest range of $\delta^2H_{wax}$ values in any given region, in line with the decrease in variability that follows an increase in integration area and integration time. Marine sediments are expected to integrate over larger areas and longer timescales than river sediments, acquiring sediments not only from near fluvial sources but also by aeolian input or coast-parallel currents (Gagosian and Peltzer, 1986; Poynter et al., 1989; Bernhardt et al., 2016). Overall, our findings support the notion of consistent transport of $\delta^2H_{wax}$ values from the continent to the adjacent marine sediments within each climate zone.

**Table 4.** Results of the Kruskal-Wallis test performed to assess differences in $\delta^2H_{wax}$ values across soils, river sediments and marine sediments.

|  | Hyperarid | | | Arid-Semiarid | | | Humid | | |
|---|---|---|---|---|---|---|---|---|---|
|  | p-value | chi-squared | df | p-value | chi-squared | df | p-value | chi-squared | df |
| **n-C29** | 0.958 | 0.003 | 1 | 0.084 | 4.948 | 2 | 0.104 | 4.528 | 2 |
| **n-C31** | 0.560 | 0.339 | 1 | 0.334 | 2.193 | 2 | 0.284 | 2.518 | 2 |

### 4.5 Implications for paleoclimate studies

The consistency between $\delta^2H_{wax}$ values in marine sediments and in the adjacent continental river sediments and soils supports their use in paleoclimatic studies along the Chilean margin. Overall, our results encourage the use of the $\delta^2H_{wax}$ as proxy to study changes in hydrological conditions along the Chilean margin.

Our results demonstrate the potential of $\delta^2H_{wax}$ as a tracer of $\delta^2H_{pre}$ in the humid to arid zones of Chile, supporting the application of leaf wax $\delta^2H$ palaeohydrology proxy in such regions. However, we also found that hyperaridity causes a strong $^2H$ enrichment (i.e. smaller apparent fractionation), and that the relationship between hydrological variables and $\varepsilon_{wax/pre}$ is non-linear. These findings are particularly relevant to the interpretation of paleoclimatic changes, because they suggest that $\delta^2H_{wax}$ is highly sensitive to the onset of strong aridity: potentially large changes in $\delta^2H_{wax}$ could be induced by small variation in the hydrological parameters. Not accounting for the non-linearity could lead to overestimation of the magnitude of the hydrological changes, as also discussed in Hou et al. (2018).



Finally, the application of *n*-alkanes as proxies for $\delta^2H_{pre}$ requires consideration of potentially differential responses of the *n*-
535 C$_{29}$ and *n*-C$_{31}$ homologues under strong aridity conditions, likely due to different vegetation sources. We find that *n*-C$_{29}$ is more sensitive to aridity and thus, in sites with high aridity shows less negative $\delta^2H_{wax}$ values relative to *n*-C$_{31}$ (Fig. 5). Similar findings are reported in previous studies (Chen et al., 2022; Wang et al., 2013; Garcin et al., 2012). We find that this difference is particularly pronounced in arid settings where herbaceous vegetation dominates. This differential sensitivity among *n*-C$_{29}$ and *n*-C$_{31}$ could be useful in paleoclimatic studies to detect the onset of high aridity, and thus could help avoid overestimating
hydrological changes. However, such an application requires additional information about the n-alkanes of the vegetation from the source areas.

## 5 Conclusions

By analysing soils, river sediments, and their marine counterparts along a strong aridity gradient in Chile we tested the robustness with which the hydrogen isotope composition of leaf waxes ($\delta^2H_{wax}$) in soils and river sediments reflect modern
climate conditions, and how accurately marine sediments preserve continental $\delta^2H_{wax}$ values. We corroborate previous findings that in humid to arid zones, $\delta^2H_{wax}$ values are largely controlled by $\delta^2H_{pre}$. However, in hyperarid zones, $\delta^2H_{wax}$ values are more strongly influenced by evapotranspirative enrichment, resulting in a non-linear relationship of $\delta^2H_{wax}$ with hydrological variables. Using established models, we show that changes in relative humidity, mean annual precipitation and actual evapotranspiration can explain most of the $^2H$ enrichment for the *n*-C$_{29}$ homologue. The *n*-C$_{31}$ homologue is less sensitive to
changes in hydrological conditions, suggesting that the $\delta^2H_{wax}$ differences between the homologues reflect differences in leaf-wax sources and their differential use of enriched leaf water and unenriched xylem water, as herbaceous vegetation is dominant under arid conditions. Further, the non-linearity that we found is particularly relevant in hyperarid zones, where small variations in hydrological parameters translate into large changes in $\delta^2H_{wax}$ values. In paleoclimate studies, care must hence be taken when interpreting records in arid to hyperarid regions as records of $\delta^2H_{pre}$ alone, as changes in $\delta^2H_{wax}$ can become
decoupled from changes in $\delta^2H_{pre}$ and are rather driven by actual evapotranspiration. The marine core-top samples that we analysed faithfully reflect continental $\delta^2H_{wax}$ values from river sediments and soils along the entire aridity gradient, showing a land-ocean connectivity and preservation of terrestrial signals in the marine realm, even under the different hydrological regimes along the Chilean coast.

## Data availability

All supplementary tables (S1–S8) and figures (S1–S4) are available in the data publication Gaviria-Lugo et al. (2023): http://doi.org/10.5880/GFZ.3.3.2023.001 (The DOI link is not active yet, temporary link provided to reviewers). The data publication is hosted by the GFZ data services. Upon final acceptance of the manuscript for publication, the DOI link will be

activated and consequently the data will be made available under the Creative Commons Attribution 4.0 International (CC BY 4.0) open-access license. When using these data please cite this paper.

## Sample availability

The metadata of all the IGSN-registered samples used for this study (samples in Table 1) can be accessed via http://igsn.org/[Insert sample IGSN number here]

## Author contribution

NG: Conceptualized the research, collected, and prepared samples, compiled and analysed data, prepared the original manuscript draft. CL: Conceptualized the research, collected samples, reviewed, and edited the manuscript. HW: Acquired funding, conceptualized the research, collected samples, reviewed, and edited the manuscript. AB: Acquired funding, conceptualized the research, collected samples, reviewed, and edited the manuscript. PF: Conceptualized the research, collected samples, reviewed, and edited the manuscript. MM: Provided marine sediment samples, reviewed, and edited the manuscript. OR: Assisted and supervised the lab work, reviewed, and edited the manuscript. DS: Supervised and conceptualized the research, acquired funding, collected samples, reviewed, and edited the manuscript.

## Competing Interest

The authors declare that they have no conflict of interest.

## Acknowledgments

We acknowledge support from the German Science Foundation (DFG) priority program SPP-1803 "EarthShape: Earth Surface Shaping by Biota". This research was supported through DFG grants BE5070/6-1 (to AB), WI3874/7-1 (to HW) and SA1889/3-1 (to DS). We are grateful to the Chilean National Park Service (CONAF) for providing access to the sample locations (Pan de Azúcar, La Campana and Nahuelbuta) and on-site support of our research. We thank Friedhelm von Blanckenburg and Todd Ehlers for initiating and leading the SPP Earthshape. We thank Kirstin Übernickel and Leandro Paulino for their support in the coordination of the EarthShape program. The marine sample material was stored and supplied by the GeoB Core Repository at the MARUM – Center for Marine Environmental Sciences, University of Bremen, Germany.

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
