# Peer review of "Climatic controls on leaf wax hydrogen isotope ratios in terrestrial and marine sediments along a hyperarid to humid gradient"

_EGUsphere, 2023_

## Author Comment (AC2)

**Response to comments from Anonymous Referee #2**

Thank you for taking the time to review our manuscript and the constructive comments. Below you find a point-by-point response to each of the comments and revisions suggested. The line numbers refer to the original preprint version that you reviewed.

- Referee comments are in black color.
- Replies to referee comments are in blue.
- **The new paragraphs, sentences or words added to the manuscript are underlined and in orange.**

The manuscript contains a dataset of $\delta^2H_{wax}$ values from different terrestrial, riverine and marine sediment samples across environments with different aridity indexes in Chile. These $\delta^2H_{wax}$ values are accompanied by $\delta^2H$ values of precipitation, and a large set of different environmental characteristics, with the aim of identifying how well sediment derived $\delta^2H_{wax}$ values track $\delta^2H$ values of precipitation, and with that provide new insights into the validity of using $\delta^2H_{wax}$ values for paleoclimatic reconstructions. The results show that on a global scale, the obtained $\delta^2H_{wax}$ values follow $\delta^2H$ values of precipitation quite well. However, within the dataset itself, across the aridity gradient, other environmental drivers also appear to become important in shaping $\delta^2H_{wax}$ values. The latter even seems to differ for $\delta^2H$ values of the two studied leaf wax *n*-alkane C-chain lengths, possibly related to changes in vegetation types. Lastly, the $\delta^2H_{wax}$ values found in marine sediment samples reflect the $\delta^2H_{wax}$ values from the terrestrial and riverine sediments.

I enjoyed reading the manuscript, which is written well with clear explanations of objectives and implications. The data presented here provides interesting new insight into how well $\delta^2H_{wax}$ values track $\delta^2H$ values of precipitation on a global scale, but also considers in more detail deviation of $\delta^2H_{wax}$ from the expected $\delta^2H_{pre}$ pattern by different drivers, like changes in evapotranspiration and vegetation type, along the aridity gradient. I think this is a valuable new approach to gain more insights into the drivers of $\delta^2H_{wax}$ values. Although I am not an expert in the modelling approach and therefore cannot judge its accuracy very well, the explanation of the model was clear enough that I could follow what was being done. I only have a few minor comments that may help further strengthen the manuscript, and I think this paper is suitable for publication after these minor points have been addressed.

Thank you for your thoughtful and positive comments on our manuscript.

**Minor comments:**

1. Although changes in $\delta^2H_{wax}$ values as an effect of differences in vegetation type are discussed in section 4.2, it is not addressed in the manuscript introduction. In L47-L59 I believe it might be valuable to already introduce the possible effects of species variation on $\delta^2H_{wax}$ along the aridity gradient where changes in plant community composition may occur. Additionally, for the discussion section 4.2, it could be considered that even within taxonomically/physiologically constrained groups like herbaceous plants or eudicots, species differences in $\varepsilon_{wax/pre}$ can still be very large (Chikaraishi et al., 2004, *Phytochem*.; Gao et al., 2014, *PLoS ONE*; He et al., 2020, *Geochim. Cosmochim. Acta*; Baan et al., 2023, *Geochim. Cosmochim. Acta*). This complicates the interpretation of the effect of the broad term 'vegetation type' changes on $\delta^2H_{wax}$ values and requires more detailed knowledge on the integration of

*n*-alkanes and their $\delta^2H$ values from different plant species into the sediment (as youstate in L540).

We appreciate the feedback provided on this point. We did not originally include text in the introduction about the effects of plant species variation along Chile, because there is limited detailed information on the plant species found along Chile. Nevertheless, this comment highlights the necessity to add context to the readers that introduces this topic. In a revised version, we have now extended the paragraph to include new sentences starting in line 52 that introduce how general changes in plant communities can cause changes in $\varepsilon_{wax/pre}$ and consequently $\delta^2H_{wax}$.

Sentences added in line 52: Yet, $\varepsilon_{wax/pre}$ values can be affected by the type of plant communities sourcing the *n*-alkanes. Generally, $\varepsilon_{wax/pre}$ values are higher in $C_3$ plants than in $C_4$ plants (Chikaraishi et al., 2004; Smith & Freeman, 2006, Kahmen et. al 2013, Sachse et al. 2010, Sachse et al. 2012). It has been suggested that these differences originate due to specific discrimination against $^2H$ between distinct photosynthetic pathways (Chikaraishi et al., 2004), as well as due to different pools of biosynthetic source waters fed by different mixtures of enriched leaf water and unenriched soil water (Kahmen et al. 2013), or due to both processes. Moreover, studies analyzing plants by growth form (Griepentrog et al., 2019; Liu et al., 2016) or even at the species level (Gao et al., 2014) show a strong control of vegetation type or species on $\varepsilon_{wax/pre}$, explained by physiological and biochemical factors that vary among different plant taxa (Gao et al., 2014; Liu et al., 2016). In addition to the vegetation effects,

Regarding the discussion section 4.2, we agree with the view expressed by the reviewer, that even within the same plant growth form (i.e., herbaceous, or woody), differences in $\varepsilon_{wax/pre}$ between different plant species can still be very large. We acknowledge that the use of plant growth forms might not be ideal over short spatial scales of meters, as it might oversimplify all the complexities introduced by individual plant species. But at the scale of our study, we believe it is currently the best approach in terms of feasibility. At the moment, to our knowledge there are not open and accurate datasets of plant species distribution at the scale of the catchments we studied in Chile. Therefore, we have made the decision to use plant growth forms instead of plant species distribution datasets because of the availability of the current remote sensing datasets. We hope that soon accurate plant species distribution become openly available. This would allow future studies to investigate the effects of individual plant species on $\delta^2H_{wax}$ values at a catchment scale, and to improve the accuracy of paleoenvironmental reconstructions.

To address the reviewer's comment, we have rearranged the paragraph between lines 439 and 453, also changed the wording of some sentences in the paragraph between lines 454 and lines 459, and finally added new sentences starting on line 452 that mention the possible effects of distinct plant species of the same growth form and the need for further research to investigate the universality of the findings discussed in section 4.2.

Sentences added in line 452: or particular plant species. In this study we focused on the level of plant growth forms due to the lack of plant species datasets at the scale of our study areas. However, it is important to note that within the same plant growth form, different plant species might exhibit distinct apparent and/or biosynthetic fractionation values and consequently affect $\delta^2H_{wax}$ differently. The influence of both plant growth

**forms and plant species on sedimentary δ²H$_{wax}$ over large spatial scales is unclear, and more studies are needed to fully understand these relationships.**

2. Even though in Fig. 2A & B all of the datapoints from the Chilean dataset would be considered to fall within error among the datapoints in the global dataset (i.e. roughly falls on the expected line in a global δ²H$_{pre}$ gradient), it seems that once the Chilean dataset is isolated, there is no longer a strong relationship between δ²H$_{wax}$ and δ²H$_{pre}$ Can the authors comment on the relation between δ²H$_{wax}$ and δ²H$_{pre}$ within the Chilean dataset?

We thank the reviewer for the thoughtful comment. We fully agree with this observation about the Chilean dataset looking disperse around the global relationship. We refer here to Table 2. There, we present the statistical parameters of the linear regressions obtained between δ²H$_{pre}$ and δ²H$_{wax}$ both for the global dataset and the isolated Chilean dataset, separating it further into only soil samples, only river samples, and soils plus rivers samples together. The results from Table 2, which are described in the manuscript, between lines 293 and 296 indicate that the relationship between δ²H$_{pre}$ and δ²H$_{wax}$ is strong and significant when considering the river samples, as well as rivers and soil samples together, but for the dataset of only soils it is not significant. Furthermore, the results indicate that for the $n$-C$_{29}$ homologue, the combined dataset of Chilean soils and rivers shows slope and intersect values (Slope = 0.76; intercept = -118) that are equivalent to the slope and intersect values from the global dataset (Slope = 0.79; intercept = -124), while all the other datasets have significant differences for the slope and intersect values relative to the global dataset.

The implications of Table 2 results are discussed in section 4.4. However, we acknowledge that Table 2 could be improved, and the results should be directly referenced in the discussion of the manuscript so that the findings are not overlooked. Therefore, we did three things, first we modified the formatting of the p-value numbers to provide more clarity. Second, we added an extra column called "significance" that represents the significance of the correlations obtained between δ²H$_{pre}$ and δ²H$_{wax}$ using asterisks symbols. If a p-value is greater than 0.05 there are no asterisks; if a p-value is between 0.05 and 0.01, it is represented with one asterisk (*), a p-value is between 0.01 and 0.001, it is represented with 2 asterisks (**) and a p-value less than 0.001 is represented with three asterisks (***). Third, we changed the topic sentence starting section 4.4 and added a new one in line 491.

Sentence added in line 491: Analysing the Chilean dataset at the differential spatial scales, we found that δ²H$_{pre}$ values showed better correlation with δ²H$_{wax}$ values from river sediments than from soils (Table 2).

From Fig. 2A & B it looks like this relationship is not very strong, and if this is the case, could this be an effect of uncertainty in δ²H$_{pre}$ values, or an effect of additional environmental/biological control on δ²H$_{wax}$ values?

We thank you for the opportunity to clarify some of the findings of our study by asking what could control δ²H$_{wax}$ values beyond δ²H$_{pre}$? And questioning if this is an effect of uncertainties in δ²H$_{pre}$ values or is due to further environmental/biological controls. In the manuscript we tackle these questions in section 4.1 and the subsections that are contained therein. The uncertainties in δ²H$_{pre}$ values are discussed in subsection 4.1.1,

where we validate the accuracy of $\delta^2H_{pre}$ values derived from the Online Isotopes in Precipitation Calculator (OIPC) by comparing them to $\delta^2H_{pre}$ measurements from the IAEA in 9 stations along Chile. The results are displayed in figures S1.A and S1.B, together with the statistical parameters of the linear regressions. The data presented provides strong evidence that the OIPC-derived $\delta^2H_{pre}$ values are accurate along the Chilean gradient. The high and significant correlation values between the predicted and the long-term measured values suggest that the OIPC model is accurately capturing the variations in $\delta^2H_{pre}$. Although there are some uncertainties associated with the OIPC model, these uncertainties do not appear to be systematic and do not bias the results towards any particular aridity zone.

With respect to the environmental controls on $\delta^2H_{wax}$ values, in the manuscript we divide this discussion in two. In subsection 4.1.2 we first discuss the controls on $\delta^2H_{wax}$ values for samples from the humid, semiarid and arid zone. In subsection 4.1.3, we discuss the controls on $\delta^2H_{wax}$ values for samples from the hyperarid zone. We followed this approach based on the residuals analysis showed in Fig. 2B and 2D and the results in Table 3. These results show that in the arid, semiarid and humid zone, the $\delta^2H_{wax}$ values do not significantly deviate from what is predicted by the global regression between $\delta^2H_{wax}$ and $\delta^2H_{pre}$. This suggest that, in these regions, variability in $\delta^2H_{wax}$ values is primarily driven by variability in $\delta^2H_{pre}$ values. However, in the hyperarid zone, residuals do significantly deviate from the global regression between $\delta^2H_{wax}$ and $\delta^2H_{pre}$, indicating that additional factors besides $\delta^2H_{pre}$ values affect $\delta^2H_{wax}$ values. We suggest that in the hyperarid zone $\delta^2H_{wax}$ values are heavily controlled by evapotranspirative processes, in addition to $\delta^2H_{pre}$ values. We reach this conclusion based on the analysis of apparent fractionation values against climatic parameters that we discuss between lines 381 and 406, as well as the mechanistic modelling of $^2H$ enrichment in soils and leaves that is discussed between lines 413 and 436.

The discussion about biological controls on $\delta^2H_{wax}$ values is presented in section 4.2. Initially we only discussed how plant growth forms could control the differences in $\delta^2H_{wax}$ values identified between the $n\text{-}C_{29}$ and $n\text{-}C_{31}$ homologues. However, after the first comment of your review, we added further sentences in line 452 to discuss the effect of plant species on $\delta^2H_{wax}$ values. The sentences added are mentioned above as the reply to the first comment.

As a result of this, what magnitude of error could be introduced when reconstructing $\delta^2H_{pre}$ from $\delta^2H_{wax}$ for a given site that may be subject to additional environmentally/ biologically induced variation in $\delta^2H_{wax}$ values over time? I suppose the latter is difficult to answer quantitatively, but perhaps the authors can comment on this.

Thank you for this comment. As you mention, the uncertainties on reconstructed $\delta^2H_{pre}$ values are hard to quantify. Nonetheless, we acknowledge that it is necessary to provide more clarity and context about these potential uncertainties to the readers. Thus, based on this and on your third comment, we have added a new paragraph after line 541 and additionally revised and modified the paragraphs between lines 524 and 541. The new paragraph as well as the new additions to the paragraphs between lines 524 and 541 are shown below as part of the response to the third comment.

Overall, I find the different comparisons made in Fig. 2 very interesting, but it might be valuable to clear up the interpretation and implications of the results on different spatial scales.

We appreciate this comment, and we also consider important to interpret and discuss the implications of the results at the level of the different spatial scales. The discussion of the interpretations and implications of this on the different spatial scales is done in section 4.4, where we added a new sentence at line 491 based on the first question of this comment, which can be seen above, we additionally modified Table 2, as explained above, to bring more clarity to the results obtained at the different spatial scales. We believe that these additions to the manuscript address both the first question (about isolating the Chilean dataset) and the last question (about the interpretations and implications on different spatial scales) of this comment.

3. L527-L533: The results presented suggest that changes in the hydrological and vegetation characteristics of a given study site over time (i.e. irrespective of its current aridity state) can introduce some error in the reconstructed $\delta^2H_{pre}$ from sedimentary $\delta^2H_{wax}$ values, which is somewhat in contrast to the statement made in L527. As such, the continuation of this paragraph seems to be slightly opposing the initial statement, as hydrological changes may not be reflected in $\delta^2H_{pre}$ Perhaps this paragraph could be slightly revised to provide a better overview of the nuances required for paleoclimate reconstructions from $\delta^2H_{wax}$ values.

Thank you for your comment. We agree that the results of our study suggest that changes in the hydrological and vegetation characteristics of an area over time can introduce additional errors in the reconstructed $\delta^2H_{pre}$ from sedimentary $\delta^2H_{wax}$ values. However, we also believe that the aridity state of an area is an important factor to consider when interpreting $\delta^2H_{wax}$ values. In our study, we found that the significant evapotranspirative effects on $\delta^2H_{wax}$ only have been identified in hyperarid zones. In contrast, the humid, semiarid, and arid zones are less affected by evapotranspirative fractionation, and $\delta^2H_{wax}$ generally reflects $\delta^2H_{pre}$ in these zones. Thus, we believe that it is important to consider the aridity state of an area when using $\delta^2H_{wax}$ values as a paleoenvironmental proxy.

To clarify this further, and as part of the response to one of the questions in your second comment above, we have revised and rephrased the paragraph starting in line 527. Additionally, we combined this paragraph with the paragraph starting in line 524, to make it more cohesive and consistent. The new combined, revised, and rephrased version of the paragraph is as follows (the changes to the text are in orange): Our results demonstrate the potential of $\delta^2H_{wax}$ as a proxy for $\delta^2H_{pre}$ in the humid to arid zones of Chile. We found that $\delta^2H_{wax}$ values in marine sediments are consistent with those in river sediments and soils from the adjacent continent, supporting the use of marine sedimentary $\delta^2H_{wax}$ as a tracer of continental $\delta^2H_{pre}$. However, our analysis also revealed that hyperaridity can cause strong $^2H$ enrichment (i.e., smaller $\varepsilon_{wax/pre}$ values) and non-linear relationships between hydrological variables and $\varepsilon_{wax/pre}$. These findings suggest that $\delta^2H_{wax}$ is highly sensitive to the onset of extreme aridity. While $\delta^2H_{wax}$ values largely reflect $\delta^2H_{pre}$ values in humid, semiarid, and arid settings, in hyperarid regions, the strong evapotranspirative effects on $\delta^2H_{wax}$ could lead to an overestimation of $\delta^2H_{pre}$ values, and consequently of hydrological changes, as also discussed in Hou et al. (2018). Because of this, it is crucial to consider the aridity states that an area may have experienced when using $\delta^2H_{wax}$ as a paleoenvironmental proxy.

We also revised and changed some wordings to the paragraph starting in line 534. The new revised version of the paragraph is as follows (the changes to the text are in orange): Furthermore, our analysis revealed differential responses of the $n$-C$_{29}$ and $n$-C$_{31}$ homologues under strong aridity conditions, likely due to different vegetation sources. We found that $n$-C$_{29}$ is more sensitive to aridity and exhibits less negative $\delta^2$H$_{wax}$ values relative to $n$-C$_{31}$ in sites with high aridity (Fig. 5). Similar findings have been reported in previous studies (Chen et al., 2022; Garcin et al., 2012; Wang et al., 2013). We found that the difference between $\delta^2$H$_{wax}$ values in $n$-C$_{29}$ and $n$-C$_{31}$ is particularly pronounced in arid settings, especially where herbaceous vegetation dominates. This differential sensitivity could be useful in detecting the onset of high aridity, and thus could help avoiding overestimation of hydrological changes. However, such an application requires additional information about the $n$-alkanes of the vegetation from the source areas.

Finally, we added a new paragraph after line 541: In summary, along the Chilean humid to arid zones, our results support the use of $\delta^2$H$_{wax}$ as a proxy for $\delta^2$H$_{pre}$ and to study changes in paleohydrological conditions. However, with the onset of hyperaridity, $\delta^2$H$_{wax}$ values can become decoupled from $\delta^2$H$_{pre}$ and be controlled by evapotranspirative processes.

**Phrasing/textual comments:**

L30: Italicize '$n$' in '$n$-alkanes'. This is not consistently done throughout the manuscript.

This was revised and fixed throughout the whole manuscript.

L52: although 'less negative' is not incorrect, I find that this can be a somewhat confusing term. More straightforward referencing between different δ and ε values could simply be 'higher' or 'lower' than (in this specific case 'higher'). Also goes for further on in the manuscript (e.g. L286 and L287).

This was revised and fixed throughout the whole manuscript.

L65: superscript of '13' and subscript of 'wax' should be fixed.

This was fixed following the same recommendation from reviewer 1.

L93: Was the internal standard also used as a recovery standard to account for losses during sample processing? This is not mentioned later in the paragraph regarding $n$-alkane quantification (L107).

Strictly, we only used the internal standard to normalize the peak areas of the n-alkanes in the chromatogram using the peak area from the standard and the known concentration of standard added. This allowed us to compare the relative abundance of the n-alkanes in different samples. We spiked the sample after extracting the total lipid extract (TLE) from the sediments, but before performing the separation of the TLE through solid phase extraction (SPE). This could be seen as a recovery standard to some researchers, but others would suggest naming it otherwise.

To strive for clarity, we added the following paragraph in line 107: We used the internal standard to normalize the peak areas of the n-alkanes in the chromatogram. The internal

**standard was spiked into the sample after extracting the TLE from the sediments, but before performing the separation of the TLE through SPE. This allowed us to compare the relative abundance of the n-alkanes in different samples, while accounting for any losses that may have occurred during SPE.**

L198: Reference format: should not be in separate brackets?

*Corrected following the same suggestion made by reviewer 1.*

L445: 'They' refers to herbaceous plants? 'Deeper rooting depths' relative to what (other vegetation types or with aridity, I presume the latter, but it is not entirely clear from this sentence)?

*Thank you for bringing this to our attention, we rephrased the sentence in line 445 to increase clarity:* **Also, in desert ecosystems, herbaceous plants generally have deeper rooting depths in comparison to woody plants like shrubs,**

L493-494: '…, as they integrate over larger regions.' seems a bit confusing at the end of the sentence since '… as they average both vegetation and climatic variability to a greater extent …' is already mentioned before (i.e. last part of the sentence is redundant I think). Perhaps change to something like: '… as they integrate both vegetation and climatic variability over larger regions than soil samples.'

*Thanks for this suggestion, it was implemented directly as suggested.*

Table 4: This table seems somewhat redundant, as the p values are already shown in Fig. 7. The table itself could perhaps be moved to a supplemental info document or somehow processed into the text, if manuscript length would be in issue.

*After revising it, we agree with this comment and see no relevant new information presented in Table 4. Since the p-values are shown in Fig. 7, it can be considered redundant, therefore we move it into the supplement of the manuscript where it can be found as Table S9.*

**References:**

Chen, G., Li, X., Tang, X., Qin, W., Liu, H., Zech, M., & Auerswald, K. (2022). Variability in

  pattern and hydrogen isotope composition ($\delta^2$H) of long-chain n-alkanes of surface

  soils and its relations to climate and vegetation characteristics: A meta-analysis.

  *Pedosphere*, 32(3), 369–380. https://doi.org/10.1016/S1002-0160(21)60080-2

Chikaraishi, Y., Naraoka, H., & Poulson, S. R. (2004). Hydrogen and carbon isotopic

  fractionations of lipid biosynthesis among terrestrial (C3, C4 and CAM) and aquatic

plants. *Phytochemistry*, *65*(10), 1369–1381.

https://doi.org/10.1016/j.phytochem.2004.03.036

Gao, L., Edwards, E. J., Zeng, Y., & Huang, Y. (2014). Major evolutionary trends in hydrogen

isotope fractionation of vascular plant leaf waxes. *PloS One*, *9*(11), e112610.

Garcin, Y., Schwab, V. F., Gleixner, G., Kahmen, A., Todou, G., Séné, O., Onana, J.-M.,

Achoundong, G., & Sachse, D. (2012). Hydrogen isotope ratios of lacustrine

sedimentary n-alkanes as proxies of tropical African hydrology: Insights from a

calibration transect across Cameroon. *Geochimica et Cosmochimica Acta*, *79*, 106–

126. https://doi.org/10.1016/j.gca.2011.11.039

Griepentrog, M., De Wispelaere, L., Bauters, M., Bodé, S., Hemp, A., Verschuren, D., &

Boeckx, P. (2019). Influence of plant growth form, habitat and season on leaf-wax nalkane hydrogen-isotopic signatures in equatorial East Africa. *Geochimica et

Cosmochimica Acta*, *263*, 122–139. https://doi.org/10.1016/j.gca.2019.08.004

Hou, J., Tian, Q., & Wang, M. (2018). Variable apparent hydrogen isotopic fractionation

between sedimentary n-alkanes and precipitation on the Tibetan Plateau. *Organic

Geochemistry*, *122*, 78–86. https://doi.org/10.1016/j.orggeochem.2018.05.011

Liu, J., Liu, W., An, Z., & Yang, H. (2016). Different hydrogen isotope fractionations during

lipid formation in higher plants: Implications for paleohydrology reconstruction at a

global scale. *Scientific Reports*, *6*(1), 19711.

Smith, F. A., & Freeman, K. H. (2006). Influence of physiology and climate on $\delta$D of leaf

wax n-alkanes from C3 and C4 grasses. *Geochimica et Cosmochimica Acta*, *70*(5),

1172–1187. https://doi.org/10.1016/j.gca.2005.11.006

Wang, Y. V., Larsen, T., Leduc, G., Andersen, N., Blanz, T., & Schneider, R. R. (2013). What

does leaf wax $\delta$D from a mixed C3/C4 vegetation region tell us? *Geochimica et

Cosmochimica Acta*, *111*, 128–139. https://doi.org/10.1016/j.gca.2012.10.016

---

## Author Response (AR1)

**List of changes to the manuscript and point-by-point response to the comments and suggestions done by reviewers**

Dear editors and reviewers, here we provide the combined response given to the comments from the reviewers 1 and 2 to our manuscript, as well as a compilation of the changes done to the manuscript.

**Combined point-by-point responses to referees.**

We follow the next convention for the point-by-point response:

- *Referee comments are in italic and black color*
- Replies to referee comments are in blue.
- **The new paragraphs, sentences or words added to the manuscript are underlined and in orange.**

**Response to comments from Anonymous Referee #1**

Thank you for your positive and constructive comments on our submitted manuscript. Below you find a point-by-point response to each of the comments and revisions suggested. The line numbers refer to the original preprint version that you reviewed.

**Overview:**

*This manuscript presents new d2Hwax data from soils, river and marine sediments along the Chilean coast. The results show that there is a constant apparent fractionation in humid regions, whereas in arid regions evapotranspiration contributes to the d2Hwax signal. The d2Hwax of C29/C31 is shown to also be related to the aridity gradient, and potentially can reflect vegetation type changes. d2Hwax of marine sediments reflect the terrestrial d2Hwax input.*

**Review:**

*This manuscript is very interesting, provides novel data and important global insights and is structured and written very well. I congratulate the authors for a well-presented paper. The MS presents novel and systematic data, combined with a wide array of global databases (climate, vegetation etc.) and an updated d2Hwax database. The manuscript is particularly interesting in its assessment of the evaporation effect on d2Hwax in arid regions, and provides a global perspective on this process. The modeling and model parametrization are explained very well and lay out the method for utilizing this method in other places.*

*I recommend publishing the paper pending some minor and textual comments.*

Thanks for the appreciation of our work.

**Minor:**

*L116. Please explain how you calculate the uncertainty of the d2H values (e.g., average of duplicates? long-term error? error of the A6? etc..).*

**Thanks for bringing this to our attention. To bring more clarity for the readers we added the following underlined sentence to the line 116 of the manuscript:** **The uncertainty of the $\delta^2H_{wax}$ values was calculated using the standard deviation between the duplicate measurements of each sample.**

*L297. This correlation is for soils and lakes combined?*

**The correlation mentioned in this line is indeed for soils and lakes combined, as it is also stated in the column 'sediment type' in table 2. To make this clearer we added a description at the table bottom to clarify what 'sediment type' refers to. Additionally added the next underlined phrase in line 297 of the manuscript:** **combining the lakes and soils data,**

*L350. Why not present the annual average from each site compared to the OIPC data? Or, maybe just average the growing season months? In addition, it would be useful if you could provide the average residuals and standard deviation of the residuals (what is the difference between measured and OIPC data in permil).*

**Thanks for commenting this and for the suggestions. We think the idea of the annual average is very good, we have implemented this and added a new analysis and respectively a figure to the supplement (Figure S1.C), where we compared the mean annual values of each GNIP site compared to the mean annual prediction of the OIPC. We also added a table (Table S2.1) in the supplement showing the average residuals and standard deviation of the residuals for each of the analysis.**

**Regarding the growing season months, we decided to not pursue this type of analysis in our study due to the added complexity and uncertainty that would come from defining accurate growing season months along our study area. It has been shown that plants along the Chilean climatic gradient present a different timing of the growing season (Hajek & Gutiérrez, 1979), and even within the same catchment areas it is expected that plants in the upper regions of the catchments will have different growing season months than plants in the lower regions (Arroyo et al., 1981). Consequently, in our study area, it is not correct to define some unique months of the year as the growing season. Instead, to pursue this type of analysis, multiple growing seasons should be defined along the gradient and even be considered inside each catchment. We understand that eventually defining multiple growing seasons would be more precise, but we believe that the added uncertainties would not be justified. Therefore, we decided to avoid these added uncertainties and used the mean annual OIPC value for our catchments and sampling sites which encompasses an average signal that would be integrated by the plants of our study areas.**

*L498 – 519. The statistical test shows that the marine, river and soils d2H overlap and are not statistically different from one another. However, Fig. 7 shows that marine sediments are, on average, heavier from rivers and soils and don't really overlap at the 1 sigma level. Is this of importance? The Peru current flows northward, so ocean mixing would cause the opposite effect. Maybe higher contribution from coastal sediments (that should be heavier than the rivers based on Fig. 1c)?*

**We thank the reviewer for the thoughtful comment. Certainly Fig. 7 visually indicates that heavier $\delta^2H_{wax}$ values were measured in marine sediments in comparison to soils and**

river sediments. We acknowledge that this may suggest that marine sediments have a different source than the soils and river sediments of their respective aridity zone. However, the statistical test performed indicates that median $\delta^2H_{wax}$ values are not statistically different among the different sediment types measured. This suggests that, although the values could be higher in marine sediments, the difference is not significant.

However, we accept that it could be a valuable insight for some readers, thus we included the following underlined paragraph discussing this observation in line 520: In Fig. 7, $\delta^2H_{wax}$ values from marine sediments generally display higher $\delta^2H_{wax}$ values in marine sediments in comparison to soils and river sediments. Although the difference between $\delta^2H_{wax}$ values among the sediment types is not statistically significant, higher $\delta^2H_{wax}$ values in marine sediments might be attributed to differences in sourcing and transport of the continental sediments. However, given the limited sample set of paired marine and river sediments in the arid region and the absence of statistically significant differences in $\delta^2H_{wax}$ values between the sediment types, we consider further discussion would be too speculative at this point.

**Textual comments:**

*L33. Add the abbreviation d2H$_{wax}$ (instead of line 37)*

As suggested, we added the abbreviation in line 33 instead of line 37.

*L65. Notation d13C X2*

The notation was accordingly corrected.

*L112. Notation H3+*

The notation was corrected.

*L181. The wording here is not so clear (what is the purpose of this test? Testing the similarity of two populations?). Can you please rephrase.*

Thanks for bringing this to our attention. The goal of the Kruskal-Wallis test is to statistically test the hypothesis that the medians of two or more groups are similar. If the p-value of the test is <0.05 this hypothesis must be rejected, indicating a statistically significant difference between the medians of the groups. Being a non-parametric method, it does not make any assumptions about the distribution of the data, an using the median helps to avoid errors induced by outliers in the data. This gives robustness to the test and makes it applicable to our case.

To add clarity in our manuscript we rephrased the paragraph starting in line 182, the new paragraph starting in line 182 is the following: The Kruskal-Wallis test was used to statistically test the hypothesis that the median $\delta^2H_{wax}$ values are similar among the different aridity zones and sediment types. This non-parametric test does not make any assumptions about the distribution of the data, and using the median instead of the mean helps to avoid errors induced by outliers in the data. If the p-value of the test is <0.05, then the null hypothesis must be rejected, indicating a statistically significant difference between the medians of the groups (Kruskal & Wallis, 1952). We performed the test using

the function *kruskal.test()* of the *stats* package version 4.2.1 from the R programming language (R Core Team, 2022).

L198. Reference format

The reference format was corrected.

L199. missing "back to isotopic ratios"

This was added as suggested.

L273. Table 1 - IGSN not defined

International Geo Sampling Number (IGSN) was defined at the bottom of the table.

Table 2. df, is this the same as the number of samples used for the regression? If so, I think number of samples is a more straightforward definition of this.

df means degrees of freedom, which is the number of independent observations (samples) minus the number of parameters estimated by the model. Since our linear regression model estimates the parameters slope and intersect, then it is the number of samples minus 2. To add clarity, we added a column called 'number of samples' and also added a definition of df at the table bottom.

L475. Should be 'explained' not 'exposed'

This was changed as suggested.

L547. Maybe 'also' instead of 'more strongly'

This was changed to 'additionally'.

**References:**

Arroyo, M. T. K., Armesto, J. J., & Villagran, C. (1981). Plant phenological patterns in the high Andean Cordillera of central Chile. *The Journal of Ecology*, 205–223.

Bernhardt, A., Hebbeln, D., Regenberg, M., Lückge, A., & Strecker, M. R. (2016). Shelfal sediment transport by an undercurrent forces turbidity-current activity during high sea level along the Chile continental margin. *Geology*, *44*(4), 295–298.

Hajek, E. R., & Gutiérrez, J. (1979). Growing seasons in Chile: Observation and prediction. *International Journal of Biometeorology*, *23*, 311–329.

Kruskal, W. H., & Wallis, W. A. (1952). Use of ranks in one-criterion variance analysis. *Journal of the American Statistical Association*, *47*(260), 583–621.

Pizarro, O., Shaffer, G., Dewitte, B., & Ramos, M. (2002). Dynamics of seasonal and interannual variability of the Peru-Chile Undercurrent. *Geophysical Research Letters*, *29*(12), 22–1.

R Core Team, R. (2022). *R: A language and environment for statistical computing*. https://www.R-project.org/.

**Response to comments from Anonymous Referee #2**

Thank you for taking the time to review our manuscript and the constructive comments. Below you find a point-by-point response to each of the comments and revisions suggested. The line numbers refer to the original preprint version that you reviewed.

*The manuscript contains a dataset of $\delta^2H_{wax}$ values from different terrestrial, riverine and marine sediment samples across environments with different aridity indexes in Chile. These $\delta^2H_{wax}$ values are accompanied by $\delta^2H$ values of precipitation, and a large set of different environmental characteristics, with the aim of identifying how well sediment derived $\delta^2H_{wax}$ values track $\delta^2H$ values of precipitation, and with that provide new insights into the validity of using $\delta^2H_{wax}$ values for paleoclimatic reconstructions. The results show that on a global scale, the obtained $\delta^2H_{wax}$ values follow $\delta^2H$ values of precipitation quite well. However, within the dataset itself, across the aridity gradient, other environmental drivers also appear to become important in shaping $\delta^2H_{wax}$ values. The latter even seems to differ for $\delta^2H$ values of the two studied leaf wax n-alkane C-chain lengths, possibly related to changes in vegetation types. Lastly, the $\delta^2H_{wax}$ values found in marine sediment samples reflect the $\delta^2H_{wax}$ values from the terrestrial and riverine sediments.*

*I enjoyed reading the manuscript, which is written well with clear explanations of objectives and implications. The data presented here provides interesting new insight into how well $\delta^2H_{wax}$ values track $\delta^2H$ values of precipitation on a global scale, but also considers in more detail deviation of $\delta^2H_{wax}$ from the expected $\delta^2H_{pre}$ pattern by different drivers, like changes in evapotranspiration and vegetation type, along the aridity gradient. I think this is a valuable new approach to gain more insights into the drivers of $\delta^2H_{wax}$ values. Although I am not an expert in the modelling approach and therefore cannot judge its accuracy very well, the explanation of the model was clear enough that I could follow what was being done. I only have a few minor comments that may help further strengthen the manuscript, and I think this paper is suitable for publication after these minor points have been addressed.*

Thank you for your thoughtful and positive comments on our manuscript.

**Minor comments:**

1. *Although changes in $\delta^2H_{wax}$ values as an effect of differences in vegetation type are discussed in section 4.2, it is not addressed in the manuscript introduction. In L47-L59 I believe it might be valuable to already introduce the possible effects of species variation on $\delta^2H_{wax}$ along the aridity gradient where changes in plant community composition may occur. Additionally, for the discussion section 4.2, it could be considered that even within taxonomically/physiologically constrained groups like herbaceous plants or eudicots, species differences in $\varepsilon_{wax/pre}$ can still be very large (Chikaraishi et al., 2004, Phytochem.; Gao et al., 2014, PLoS ONE; He et al., 2020, Geochim. Cosmochim. Acta; Baan et al., 2023, Geochim. Cosmochim. Acta). This complicates the interpretation of the effect of the broad term 'vegetation type' changes on $\delta^2H_{wax}$ values and requires more detailed knowledge on the integration of n-alkanes and their $\delta^2H$ values from different plant species into the sediment (as youstate in L540).*

We appreciate the feedback provided on this point. We did not include text in the introduction about the effects of plant species variation along Chile, because there is no such precise information for the plant species found along Chile. But, after your comment we also see the necessity to bring context to the readers introducing this topic. Thus, we now extended the paragraph and added new sentences starting in line 52 that introduce how general changes in plant communities can cause changes in $\varepsilon_{wax/pre}$ and consequently on $\delta^2H_{wax}$.

Sentences added in line 52: Yet, $\varepsilon_{wax/pre}$ values can be affected by the type of plant communities sourcing the *n*-alkanes. Generally, $\varepsilon_{wax/pre}$ values are higher in $C_3$ plants than in $C_4$ plants (Chikaraishi et al., 2004; Smith & Freeman, 2006, Kahmen et. al 2013, Sachse et al. 2010, Sachse et al. 2012). It has been suggested that these differences originate due to specific discrimination in $^2H$ among distinct photosynthetic pathways (Chikaraishi et al., 2004), as well as due to different pools of biosynthetic source waters fed by different mixtures of enriched leaf water and unenriched soil water (Kahmen et al. 2013), or due to both processes. Moreover, studies analyzing plants by growth form (Griepentrog et al., 2019; Liu et al., 2016) or even at the species level (Gao et al., 2014) have shown a strong control of vegetation type or species on $\varepsilon_{wax/pre}$, explained by physiological and biochemical factors that vary among different plant taxonomies (Gao et al., 2014; Liu et al., 2016). In addition to the vegetation effects,

About the discussion section 4.2 we agree with the expressed by the reviewer, that even within the same plant growth form (i.e., herbaceous, or woody), differences in $\varepsilon_{wax/pre}$ between different plant species can still be very large. We acknowledge that the use of plant growth forms might not be ideal over short spatial scales of meters, as it might oversimplify all the complexities introduced by individual plant species. But at the scale of our study, we believe it is currently the best approach in terms of feasibility. At the moment, to our knowledge there are not open and accurate datasets of plant species distribution at the scale of the catchments we studied in Chile. Therefore, we have made the decision to use plant growth forms instead of plant species distribution datasets because of the availability of the current remote sensing datasets. We hope that soon accurate plant species distribution become openly available. This would allow future studies to investigate the effects of individual plant species on $\delta^2H_{wax}$ values at a catchment scale, and to improve the accuracy of paleoenvironmental reconstructions.

To address the reviewer's comment, we have rearranged the paragraph between lines 439 and 453, also changed the wording of some sentences in the paragraph between lines 454 and lines 459, and finally added new sentences starting on line 452 that mention the possible effects of distinct plant species of the same growth form and the need for further research to investigate the universality of the findings discussed in section 4.2.

Sentences added in line 452: or particular plant species. In this study we focused on the level of plant growth forms due to the lack of plant species datasets at the scale of our study areas. However, it is important to note that within the same plant growth form, different plant species might exhibit distinct apparent and/or biosynthetic fractionation values and consequently affect $\delta^2H_{wax}$ differently. The influence of both plant growth forms and plant species on sedimentary $\delta^2H_{wax}$ over large spatial scales is unclear, and more studies are needed to fully understand these relationships.

2. *Even though in Fig. 2A & B all of the datapoints from the Chilean dataset would be considered to fall within error among the datapoints in the global dataset (i.e. roughly falls on the expected line in a global $\delta^2H_{pre}$ gradient), it seems that once the Chilean dataset is isolated, there is no longer a strong relationship between $\delta^2H_{wax}$ and $\delta^2H_{pre}$ Can the authors comment on the relation between $\delta^2H_{wax}$ and $\delta^2H_{pre}$ within the Chilean dataset?*

We thank the reviewer for the thoughtful comment. We fully agree with this observation about the Chilean dataset looking disperse around the global relationship. We refer here to Table 2. There, we present the statistical parameters of the linear regressions obtained between $\delta^2H_{pre}$ and $\delta^2H_{wax}$ both for the global dataset and the isolated Chilean dataset, separating it further into only soil samples, only river samples, and soils plus rivers samples together. The results from Table 2, which are described in the manuscript, between lines 293 and 296 indicate that the relationship between $\delta^2H_{pre}$ and $\delta^2H_{wax}$ is strong and significant when considering the river samples, as well as rivers and soil samples together, but for the dataset of only soils it is not significant. Furthermore, the results indicate that for the *n*-$C_{29}$ homologue, the combined dataset of Chilean soils and rivers shows slope and intersect values (Slope = 0.76; intercept = -118) that are equivalent to the slope and intersect values from the global dataset (Slope = 0.79; intercept = -124), while all the other datasets have significant differences on slope and intersect with respect to the global dataset.

The implications of Table 2 results are discussed in section 4.4. However, we acknowledge that Table 2 could be improved, and the results should be directly referenced in the discussion of the manuscript, so that the findings are not overlooked. Therefore, we did three things, first we modified the formatting of the p-value numbers to bring more clarity. Second, we added an extra column called "significance" that represents the significance of the correlations obtained between $\delta^2H_{pre}$ and $\delta^2H_{wax}$ using asterisks symbols. If a p-value is greater than 0.05 there are not asterisks. If a p-value is between 0.05 and 0.01, it is represented with one asterisk (*). If a p-value is between 0.01 and 0.001, it is represented with 2 asterisks (**). If a p-value is less than 0.001, it is represented with three asterisks (***). Third, we changed the topic sentence starting section 4.4 and added a new one in line 491.

**Sentence added in line 491:** Analysing the Chilean dataset at the differential spatial scales, we found that $\delta^2H_{pre}$ values showed better correlation with $\delta^2H_{wax}$ values from river sediments than from soils (Table 2).

> *From Fig. 2A & B it looks like this relationship is not very strong, and if this is the case, could this be an effect of uncertainty in $\delta^2H_{pre}$ values, or an effect of additional environmental/biological control on $\delta^2H_{wax}$ values?*

We thank you for the opportunity to clarify some of the findings of our study by asking what could control $\delta^2H_{wax}$ values beyond $\delta^2H_{pre}$? And questioning if this is an effect of uncertainties in $\delta^2H_{pre}$ values or is due to further environmental/biological controls. In the manuscript we tackle these questions in section 4.1 and the subsections that are contained therein. The uncertainties in $\delta^2H_{pre}$ values are discussed in subsection 4.1.1, where we validate the accuracy of $\delta^2H_{pre}$ values derived from the Online Isotopes in Precipitation Calculator (OIPC) by comparing them to $\delta^2H_{pre}$ measurements from the IAEA in 9 stations along Chile. The results are displayed in figures S1.A and S1.B, together with the statistical parameters of the linear regressions. The data presented provides strong evidence that the OIPC-derived $\delta^2H_{pre}$ values are consistent along the Chilean gradient. The high and significant correlation values between the predicted and the long-term measured values suggest that the OIPC model is accurately capturing the variations in $\delta^2H_{pre.}$ Although there are some uncertainties associated with the OIPC model, these uncertainties do not appear to be systematic and do not bias the results towards any particular aridity zone.

With respect to the environmental controls on $\delta^2H_{wax}$ values, in the manuscript we divide this discussion in two. In subsection 4.1.2 we first discuss the controls on $\delta^2H_{wax}$ values for samples from the humid, semiarid and arid zone. In subsection 4.1.3, we discuss the controls on $\delta^2H_{wax}$ values for samples from the hyperarid zone. We followed this approach based on the residuals analysis showed in Fig. 2B and 2D and the results in Table 3. These results show that in the arid, semiarid and humid zone, the $\delta^2H_{wax}$ values are not significantly deviated from what is predicted by the global regression between $\delta^2H_{wax}$ and $\delta^2H_{pre}$. This suggest that, in these regions, variability in $\delta^2H_{wax}$ values is primarily driven by variability in $\delta^2H_{pre}$ values. However, in the hyperarid zone, residuals are significantly deviated from the global regression between $\delta^2H_{wax}$ and $\delta^2H_{pre}$, indicating that additional factors besides $\delta^2H_{pre}$ values affect $\delta^2H_{wax}$ values. We suggest that in the hyperarid zone $\delta^2H_{wax}$ values are heavily controlled by evapotranspirative processes, in addition to $\delta^2H_{pre}$ values. We reach this conclusion based on the analysis of apparent fractionation values against climatic parameters that we discuss between lines 381 and 406, as well as the mechanistic modelling of $^2H$ enrichment in soils and leaves that is discussed between lines 413 and 436.

The discussion about biological controls on $\delta^2H_{wax}$ values is presented in section 4.2. Initially we only discussed how plant growth forms could control the differences in $\delta^2H_{wax}$ values identified between the $n$-C$_{29}$ and $n$-C$_{31}$ homologues. However, after the first comment of your review, we added further sentences in line 452 to discuss the effect of plant species on $\delta^2H_{wax}$ values. The sentences added are mentioned above as the reply to the first comment.

> *As a result of this, what magnitude of error could be introduced when reconstructing $\delta^2H_{pre}$ from $\delta^2H_{wax}$ for a given site that may be subject to additional environmentally/*

*biologically induced variation in $\delta^2 H_{wax}$ values over time? I suppose the latter is difficult to answer quantitatively, but perhaps the authors can comment on this.*

**Thanks for commenting this. As you mention, the uncertainties on reconstructed $\delta^2 H_{pre}$ values are hard to quantify. Nonetheless, we acknowledge that it is necessary to bring more clarity and context about these potential uncertainties to the readers. Thus, based on this and on your third comment, we added a new paragraph after line 541 and additionally revised and modified the paragraphs between lines 524 and 541. The new paragraph as well as the new additions to the paragraphs between lines 524 and 541 are shown below as part of the response to the third comment.**

*Overall, I find the different comparisons made in Fig. 2 very interesting, but it might be valuable to clear up the interpretation and implications of the results on different spatial scales.*

**We appreciate this comment, and we also consider important to interpret and discuss the implications of the results at the level of the different spatial scales. The discussion of the interpretations and implications of this on the different spatial scales is done in section 4.4, where we added a new sentence in line 491 based on the first question of this comment, which can be seen above, we additionally modified Table 2, as explained above, to bring more clarity to the results obtained at the different spatial scales. We consider these additions to the manuscript address both the first question of this second comment about isolating the Chilean dataset, and the last question of this second comment about the interpretations and implications on different spatial scales.**

3. *L527-L533: The results presented suggest that changes in the hydrological and vegetation characteristics of a given study site over time (i.e. irrespective of its current aridity state) can introduce some error in the reconstructed $\delta^2 H_{pre}$ from sedimentary $\delta^2 H_{wax}$ values, which is somewhat in contrast to the statement made in L527. As such, the continuation of this paragraph seems to be slightly opposing the initial statement, as hydrological changes may not be reflected in $\delta^2 H_{pre}$ Perhaps this paragraph could be slightly revised to provide a better overview of the nuances required for paleoclimate reconstructions from $\delta^2 H_{wax}$ values.*

**Thank you for your comment. We agree that the results of our study suggest that changes in the hydrological and vegetation characteristics of an area over time can introduce additional errors in the reconstructed $\delta^2 H_{pre}$ from sedimentary $\delta^2 H_{wax}$ values. However, we also believe that the aridity state of an area is an important factor to consider when interpreting $\delta^2 H_{wax}$ values. In our study, we found that the significant evapotranspirative effects on $\delta^2 H_{wax}$ only have been identified in hyperarid zones. In contrast, the humid, semiarid, and arid zones are less affected by evapotranspirative fractionation, and $\delta^2 H_{wax}$ generally reflects $\delta^2 H_{pre}$ in these zones. Thus, we believe that it is important to consider the aridity state of an area when using $\delta^2 H_{wax}$ values as a paleoenvironmental proxy.**

**To clarify this further, and as part of the response to one of the questions in your second comment above, we have revised and rephrased the paragraph starting in line 527. Additionally, we combined this paragraph with the paragraph starting in line 524, to make it more cohesive and consistent. The new combined, revised, and rephrased version of the paragraph is the next (the changes to the text are in orange): Our results demonstrate the potential of $\delta^2 H_{wax}$ as a proxy for $\delta^2 H_{pre}$ in the humid to arid zones of**

Chile. We found that $\delta^2H_{wax}$ values in marine sediments are consistent with those in river sediments and soils from the adjacent continent, supporting the use of marine sedimentary $\delta^2H_{wax}$ as a tracer of continental $\delta^2H_{pre}$. However, our analysis also revealed that hyperaridity can cause strong $^2H$ enrichment (i.e., smaller $\varepsilon_{wax/pre}$ values) and non-linear relationships between hydrological variables and $\varepsilon_{wax/pre}$. These findings suggest that $\delta^2H_{wax}$ is highly sensitive to the onset of extreme aridity. While $\delta^2H_{wax}$ values largely reflect $\delta^2H_{pre}$ values in humid, semiarid, and arid settings. In hyperarid regions, the strong evapotranspirative effects on $\delta^2H_{wax}$ could lead to an overestimation of $\delta^2H_{pre}$ values, and consequently of hydrological changes, as also discussed in Hou et al. (2018). Because of this, it is crucial to consider the aridity states that an area may have experienced when using $\delta^2H_{wax}$ as a paleoenvironmental proxy.

We also revised and changed some wordings to the paragraph starting in line 534. The new revised version of the paragraph is the next (the changes to the text are in orange): Furthermore, our analysis revealed differential responses of the $n$-C$_{29}$ and $n$-C$_{31}$ homologues under strong aridity conditions, likely due to different vegetation sources. We found that $n$-C$_{29}$ is more sensitive to aridity and exhibits less negative $\delta^2H_{wax}$ values relative to $n$-C$_{31}$ in sites with high aridity (Fig. 5). Similar findings have been reported in previous studies (Chen et al., 2022; Garcin et al., 2012; Wang et al., 2013). We found that the difference between $\delta^2H_{wax}$ values in $n$-C$_{29}$ and $n$-C$_{31}$ is particularly pronounced in arid settings, especially where herbaceous vegetation dominates. This differential sensitivity could be useful in detecting the onset of high aridity, and thus could help avoiding overestimation of hydrological changes. However, such an application requires additional information about the $n$-alkanes of the vegetation from the source areas.

Finally, we added a totally new paragraph after line 541, the new paragraph added is the next: In summary, along the Chilean humid to arid zones, our results support the use of $\delta^2H_{wax}$ as a proxy for $\delta^2H_{pre}$ and to study changes in paleohydrological conditions. However, with the onset of hyperaridity, $\delta^2H_{wax}$ values can become decoupled from $\delta^2H_{pre}$ and be controlled by evapotranspirative processes.

***Phrasing/textual comments:***

*L30: Italicize 'n' in 'n-alkanes'. This is not consistently done throughout the manuscript.*

This was revised and fixed throughout the whole manuscript.

*L52: although 'less negative' is not incorrect, I find that this can be a somewhat confusing term. More straightforward referencing between different δ and ε values could simply be 'higher' or 'lower' than (in this specific case 'higher'). Also goes for further on in the manuscript (e.g. L286 and L287).*

This was revised and fixed throughout the whole manuscript.

*L65: superscript of '13' and subscript of 'wax' should be fixed.*

This was fixed following the same recommendation from reviewer 1.

*L93: Was the internal standard also used as a recovery standard to account for losses during sample processing? This is not mentioned later in the paragraph regarding n-alkane quantification (L107).*

**Strictly, we only used the internal standard to normalize the peak areas of the n-alkanes in the chromatogram using the peak area from the standard and the known concentration of standard added. This allowed us to compare the relative abundance of the n-alkanes in different samples. We spiked the sample after extracting the total lipid extract (TLE) from the sediments, but before performing the separation of the TLE through solid phase extraction (SPE). This could be seen as a recovery standard to some researchers, but others would suggest naming it otherwise.**

**To strive for clarity, we added the following paragraph in line 107:** We used the internal standard to normalize the peak areas of the n-alkanes in the chromatogram. The internal standard was spiked into the sample after extracting the TLE from the sediments, but before performing the separation of the TLE through SPE. This allowed us to compare the relative abundance of the n-alkanes in different samples, while accounting for any losses that may have occurred during SPE.

*L198: Reference format: should not be in separate brackets?*

**Corrected following the same suggestion made by reviewer 1.**

*L445: 'They' refers to herbaceous plants? 'Deeper rooting depths' relative to what (other vegetation types or with aridity, I presume the latter, but it is not entirely clear from this sentence)?*

**Thanks for bringing this to our attention, we rephrased the sentence in line 445 to increase clarity and now it is the next underlined sentence:** Also, in desert ecosystems, herbaceous plants generally have deeper rooting depths in comparison to woody plants like shrubs,

*L493-494: '..., as they integrate over larger regions.' seems a bit confusing at the end of the sentence since '... as they average both vegetation and climatic variability to a greater extent ...' is already mentioned before (i.e. last part of the sentence is redundant I think). Perhaps change to something like: '... as they integrate both vegetation and climatic variability over larger regions than soil samples.'*

**Thanks for this suggestion, it was implemented directly as suggested.**

*Table 4: This table seems somewhat redundant, as the p values are already shown in Fig. 7. The table itself could perhaps be moved to a supplemental info document or somehow processed into the text, if manuscript length would be in issue.*

**After revising it, we agree with this comment and see no relevant new information presented in Table 4. Since the p-values are shown in Fig. 7, it can be considered redundant, therefore we move it into the supplement of the manuscript where it can be found as Table S9.**

**References:**

Chen, G., Li, X., Tang, X., Qin, W., Liu, H., Zech, M., & Auerswald, K. (2022). Variability in pattern and hydrogen isotope composition ($\delta^2$H) of long-chain n-alkanes of surface soils and its relations to climate and vegetation characteristics: A meta-analysis. *Pedosphere*, *32*(3), 369–380. https://doi.org/10.1016/S1002-0160(21)60080-2

Chikaraishi, Y., Naraoka, H., & Poulson, S. R. (2004). Hydrogen and carbon isotopic fractionations of lipid biosynthesis among terrestrial (C3, C4 and CAM) and aquatic plants. *Phytochemistry*, *65*(10), 1369–1381. https://doi.org/10.1016/j.phytochem.2004.03.036

Gao, L., Edwards, E. J., Zeng, Y., & Huang, Y. (2014). Major evolutionary trends in hydrogen isotope fractionation of vascular plant leaf waxes. *PloS One*, *9*(11), e112610.

Garcin, Y., Schwab, V. F., Gleixner, G., Kahmen, A., Todou, G., Séné, O., Onana, J.-M., Achoundong, G., & Sachse, D. (2012). Hydrogen isotope ratios of lacustrine sedimentary n-alkanes as proxies of tropical African hydrology: Insights from a calibration transect across Cameroon. *Geochimica et Cosmochimica Acta*, *79*, 106–126. https://doi.org/10.1016/j.gca.2011.11.039

Griepentrog, M., De Wispelaere, L., Bauters, M., Bodé, S., Hemp, A., Verschuren, D., & Boeckx, P. (2019). Influence of plant growth form, habitat and season on leaf-wax n-alkane hydrogen-isotopic signatures in equatorial East Africa. *Geochimica et Cosmochimica Acta*, *263*, 122–139. https://doi.org/10.1016/j.gca.2019.08.004

Hou, J., Tian, Q., & Wang, M. (2018). Variable apparent hydrogen isotopic fractionation between sedimentary n-alkanes and precipitation on the Tibetan Plateau. *Organic Geochemistry*, *122*, 78–86. https://doi.org/10.1016/j.orggeochem.2018.05.011

Liu, J., Liu, W., An, Z., & Yang, H. (2016). Different hydrogen isotope fractionations during lipid formation in higher plants: Implications for paleohydrology reconstruction at a global scale. *Scientific Reports*, *6*(1), 19711.

Smith, F. A., & Freeman, K. H. (2006). Influence of physiology and climate on δD of leaf

   wax n-alkanes from C3 and C4 grasses. *Geochimica et Cosmochimica Acta*, *70*(5),

   1172–1187. https://doi.org/10.1016/j.gca.2005.11.006

Wang, Y. V., Larsen, T., Leduc, G., Andersen, N., Blanz, T., & Schneider, R. R. (2013). What

   does leaf wax δD from a mixed C3/C4 vegetation region tell us? *Geochimica et*

   *Cosmochimica Acta*, *111*, 128–139. https://doi.org/10.1016/j.gca.2012.10.016

**Compilation of changes done to the manuscript**

In the following section we list the major changes done to the manuscript because of the response to the referees' comments. The line numbers given in this section refer to the revised version (with accepted changes) of the manuscript. We follow the next convention for the changes in the manuscript:

- Additions to the manuscript are in green.
- Deleted parts of the manuscript are in red and .

1) We addressed all the problems with the notation that were present throughout the manuscript.
2) Between lines 52 and 60 we added the next sentences: Yet, εwax/pre values can be affected by the type of plant communities sourcing the *n*-alkanes. Generally, εwax/pre values are higher in C3 plants than in C4 plants (Chikaraishi et al., 2004; Smith & Freeman, 2006, Kahmen et. al 2013, Sachse et al. 2010, Sachse et al. 2012). It has been suggested that these differences originate due to specific discrimination in 2H among distinct photosynthetic pathways (Chikaraishi et al., 2004), as well as due to different pools of biosynthetic source waters fed by different mixtures of enriched leaf water and unenriched soil water (Kahmen et al. 2013), or due to both processes. Moreover, studies analyzing plants by growth form (Griepentrog et al., 2019; Liu et al., 2016) or even at the species level (Gao et al., 2014) have shown a strong control of vegetation type or species on εwax/pre, explained by physiological and biochemical factors that vary among different plant taxonomies (Gao et al., 2014; Liu et al., 2016). In addition to the vegetation effects,
3) We modified the expressions less negative and more negative referring to the isotope ratio values and changed them for higher and lower all over the manuscript.
4) Between lines 113 and 118 we added the next sentences: We used the internal standard to normalize the peak areas of the n-alkanes in the chromatogram. The internal standard was spiked into the sample after extracting the TLE from the sediments, but before performing the separation of the TLE through SPE. This allowed us to compare the relative abundance of the n-alkanes in different samples, while accounting for any losses that may have occurred during SPE. The uncertainty of the δ2Hwax values was calculated using the standard deviation between the duplicate measurements of each sample.

5) Between lines 191 and 196 we added and deleted the next sentences: The Kruskal-Wallis test was used to statistically test the hypothesis that the median δ2Hwax values are similar among the different aridity zones and sediment types. This non-parametric test does not make any assumptions about the distribution of the data, and using the median instead of the mean helps to avoid errors induced by outliers in the data. If the p-value of the test is <0.05, then the null hypothesis must be rejected, indicating a statistically significant difference between the medians of the groups (Kruskal & Wallis, 1952). ~~We used the Kruskal-Wallis test to investigate the statistical significance of differences in the median values between independent categorical groups. We explored the significance of difference in δ2Hwax values among the aridity zones, and among the different sediment types analyzed in each aridity zone. This robust non-parametric test allows for comparison of groups with dissimilar variances and does not require normally distributed data within groups.~~

6) We modified the Table 2 and added new columns to clarify details of the data presented.

7) Between lines 452 and 453 we added and deleted the next sentences: Also, in desert ecosystems, herbaceous plants generally have deeper rooting depths in comparison to woody plants like shrubs,

8) Between lines 459 and 465 we added and deleted the next: Thus, it is possible that  differences in δ2Hwax values of *n*-C31 and *n*-C29 homologues  are the manifestation of distinct physiological and biochemical characteristics among different plant growth forms or particular plant species. In this study we focused on the level of plant growth forms due to the lack of plant species datasets at the scale of our study areas. However, it is important to note that within the same plant growth form, different plant species might exhibit distinct biosynthetic fractionation values and consequently affect δ2Hwax differently. The influence of both plant growth forms and plant species on sedimentary δ2Hwax over large scales is unclear, and more studies are needed to fully understand these relationships.

9) In line 467 we changed  for These.

10) In line 468 we added potentially.

11) Between lines 472-473 we added: or possibly the presence of particular herbaceous species,

12) In line 489 we changed  for explained.

13) Between lines 506 and 507 we added and deleted the next Analysing the Chilean dataset at the differential spatial scales, we found that δ2Hpre values showed better correlation with δ2Hwax values from river sediments than from soils (Table 2).

14) In line 509 we changed  for integrate and  for to a greater extent.

15) In line 509 we deleted

16) Between lines 534-539 we added the next: In Fig. 7, δ2Hwax values from marine sediments generally display higher δ2Hwax values in marine sediments in comparison to soils and river sediments. Although the difference between δ2Hwax values among the sediment types is not statistically significant, higher δ2Hwax values in marine sediments might be attributed to differences in sourcing and transport of the continental sediments. However, given the limited sample set of paired marine and river sediments in the arid region and the absence of statistically significant differences in δ2Hwax values between the sediment types, we consider further discussion would be too speculative at this point.

17) We moved Table 4 into the supplement as table S9.

18) Between lines 541-556 we rephrased several sentences, and added new words as indicated by the following green marked words: Our results demonstrate the potential of δ2Hwax as a proxy for δ2Hpre in the humid to arid zones of Chile. We found that δ2Hwax values in marine sediments are consistent with those in river sediments and soils from the adjacent continent, supporting the use of marine sedimentary δ2Hwax as a tracer of continental δ2Hpre. However, our analysis also revealed that hyperaridity can cause strong 2H enrichment (i.e., smaller ewax/pre values) and non- linear relationships between hydrological variables and ewax/pre. These findings suggest that δ2Hwax is highly sensitive to the onset of extreme aridity. While δ2Hwax values largely reflect δ2Hpre values in humid, semiarid, and arid settings. In hyperarid regions, the strong evapotranspirative effects on δ2Hwax could lead to an overestimation of δ2Hpre values, and consequently of hydrological changes, as also discussed in Hou et al. (2018). Because of this, it is crucial to consider the aridity states that an area may have experienced when using δ2Hwax as a paleoenvironmental proxy.

Furthermore, our analysis revealed differential responses of the n-C29 and n-C31 homologues under strong aridity conditions, likely due to different vegetation sources. We found that n-C29 is more sensitive to aridity and exhibits less negative δ2Hwax values relative to n-C31 in sites with high aridity (Fig. 5). Similar findings have been reported in previous studies (Chen et al., 2022; Garcin et al., 2012; Wang et al., 2013). We found that the difference between δ2Hwax values in n-C29 and n-C$_{31}$ is particularly pronounced in arid settings, especially where herbaceous vegetation dominates. This differential sensitivity could be useful in detecting the onset of high aridity, and thus could help avoiding overestimation of hydrological changes. However, such an application requires additional information about the *n*-alkanes of the vegetation from the source areas.

19) Between lines 557-559 we added the next: In summary, along the Chilean humid to arid zones, our results support the use of δ2Hwax as a proxy for δ2Hpre and to study changes in paleohydrological conditions. However, with the onset of hyperaridity, δ2Hwax values can become decoupled from δ2Hpre and be controlled by evapotranspirative processes.

20) In line 565 we changed more strongly for additionally.

21) Between lines 604-605 we added the next: We would like to thank the two anonymous referees for their valuable comments and contributions, which have improved the quality of this paper.